# Microbial metabolite sensor GPR43 controls severity of experimental GVHD

Hideaki Fujiwara[1], Melissa D. Docampo[2], Mary Riwes[1], Daniel Peltier[3], Tomomi Toubai[1], Israel Henig[1], S. Julia Wu[1], Stephanie Kim[1], Austin Taylor[1], Stuart Brabbs[1], Chen Liu[4], Cynthia Zajac [1], Katherine Oravecz-Wilson[1], Yaping Sun[1], Gabriel Núñez[5], John E. Levine[6], Marcel R.M. van den Brink[2], James L.M. Ferrara[6] & Pavan Reddy[1]

Microbiome-derived metabolites influence intestinal homeostasis and regulate graft-versus-host disease (GVHD), but the molecular mechanisms remain unknown. Here we show the metabolite sensor G-protein-coupled receptor 43 (GPR43) is important for attenuation of gastrointestinal GVHD in multiple clinically relevant murine models. GPR43 is critical for the protective effects of short-chain fatty acids (SCFAs), butyrate and propionate. Increased severity of GVHD in the absence of GPR43 is not due to baseline differences in the endogenous microbiota of the hosts. We confirm the ability of microbiome-derived metabolites to reduce GVHD by several methods, including co-housing, antibiotic treatment, and administration of exogenous SCFAs. The GVHD protective effect of SCFAs requires GPR43-mediated ERK phosphorylation and activation of the NLRP3 inflammasome in non-hematopoietic target tissues of the host. These data provide insight into mechanisms of microbial metabolite-mediated protection of target tissues from the damage caused allogeneic T cells.

[1] Department of Internal Medicine, Division of Hematology and Oncology, University of Michigan Comprehensive Cancer Center, Ann Arbor 48109 MI, USA. [2] Department of Immunology, Memorial Sloan Kettering Cancer Center, New York 10065 NY, USA. [3] Division of Hematology and Oncology, Department of Pediatrics, University of Michigan, Ann Arbor 48109 MI, USA. [4] Department of Pathology and Laboratory Medicine, Rutgers-Robert Wood Johnson Medical School, New Brunswick 08903 NJ, USA. [5] Department of Pathology and Comprehensive Cancer Center, University of Michigan Medical School, Ann Arbor 48109 MI, USA. [6] Tisch Cancer Institute, the Icahn School of Medicine at Mount Sinai, New York 10029 NY, USA. Correspondence and requests for materials should be addressed to P.R. (email: reddypr@med.umich.edu)

Trillions of diverse bacteria colonize the intestinal tract and constitute what is termed the intestinal "microbiota"[1]. Most are nonpathogenic anaerobic commensal bacteria that contribute to host health and immune homeostasis; however, alteration of microbiota composition is affected by various factors such as foods and antibiotics, and contributes to many diseases[2]. Allogeneic bone marrow transplantation (allo-BMT) is a powerful curative therapy for hematological malignancies that causes drastic changes in the microbiota due to multiple factors including the underlying diseases, conditioning therapies, infections, antibiotic treatments, and immunologically mediated graft-versus-host disease (GVHD)[3]. Emerging data suggest that GVHD is influenced by alterations in the intestinal microbiota[1,4–7]. Disruption of microbiota composition, especially loss of specific Clostridial species and an increase of Lactobacillales or Enterobacteriales, correlates with GVHD. The critical molecular mechanisms that mediate the interaction between the microbiota and GVHD severity have yet to be elucidated[8–12].

Recently, our group reported that decreases in microbial metabolites such as short-chain fatty acids (SCFAs), which are produced by intestinal microbial fermentation of resistant fiber are associated with increased mortality from GVHD following experimental allo-BMT[13]. Oral replacement of the SCFA butyrate ameliorated GVHD mortality and was associated with better intestinal epithelial cell (IEC) repair and homeostasis. These data suggest that intestinal metabolites such as SCFA butyrate play a protective role in GVHD. In the gut, SCFAs are detected by G-protein coupled receptors (GPRs)[14], but their role in GVHD is unknown.

In this study we show that sensing of the microbiome derived SCFAs, butyrate, and propionate, by GPR43 on IECs mitigates GVHD by activating NLRP3 inflammasome via ERK phosphorylation.

## Results

**Metabolite sensor GPR43 regulates GVHD severity.** Microbial-derived SCFAs are altered following allogeneic BMT[13], but the key SCFA sensors in the gastrointestinal (GI) tract are not yet characterized. We therefore first examined the expression of known SCFA receptors such as the GPRs on IECs utilizing a well-characterized MHC disparate murine GVHD model (BALB/c → C57BL/6). IECs (CD326$^+$ cells) were harvested on day +7 and +14 after BMT and analyzed for expression of *Gpr41*, *Gpr43*, and *Gpr109a*. Allogeneic recipients demonstrated a significantly reduced expression of only *Gpr43* when compared with syngeneic recipients when normalized to actin expression and to total IEC cell numbers (Fig. 1a–c, Supplementary Fig. 1A).

In light of the differential expression of *Gpr43*, we next explored its functional relevance to GVHD using complementary, but distinct chemical and genetic approaches. Allogeneic B6 wild-type (WT) animals were orally gavaged with a GPR43 antagonist (GLPG0974, 10 mg kg$^{-1}$ per day) or diluent from day 0 to day 21[15]. Antagonist-treated allogeneic animals demonstrated significantly greater GVHD than vehicle-treated animals (Fig. 1d, e). We next tested and used both WT and *Gpr43$^{-/-}$* B6 littermate animals as BMT recipients. The allogeneic B6 WT recipients demonstrated reduced severity of GVHD when compared to the littermate B6 *Gpr43$^{-/-}$* recipients (Fig. 1f–h). All syngeneic WT and *Gpr43$^{-/-}$* B6 animals survived. The increase in GVHD severity was confirmed by histopathological analyses if the GI tract (Fig. 1i, j).

To eliminate donor/recipient strain artifact, we confirmed the protective effect of GPR43 using MHC-matched but multiple minor antigens mismatched LP/J into C57BL/6 model (Fig. 1k, l).

To eliminate microbiome artifacts related to a specific facility or colony of mice, we repeated these experiments utilizing different donor/recipient strain combinations in a second institution (Memorial Sloan Kettering Cancer Center; MSKCC). B6 WT and *Gpr43$^{-/-}$* mice were used as recipients of MHC matched minor antigens mismatched 129S1 donors and followed for survival. Once again, allogeneic B6 *Gpr43$^{-/-}$* animals demonstrated more severe GVHD than WT mice (Fig. 1m, n, $P < 0.001$) confirming that GPR43 expression in hosts mitigates predominantly GI GVHD.

**Role of microbiome in regulation of GVHD by GPR43.** Because changes in microbiome may affect GVHD mortality[6,16], we next analyzed whether the deficiency of GPR43 in the host animals altered the composition of the indigenous GI microbiota. We characterized the microbiota in feces collected from non-littermate WT B6 and *Gpr43$^{-/-}$* animals by *16S* rRNA gene sequence analysis in the animals housed at one facility (UofM). *Gpr43$^{-/-}$* B6 animals showed modest shifts in the microbiota composition when compared to B6 WT animals at baseline before BMT with the appearance of the phylum Proteobacteria, increased representation of *Prevotellaceae* and decreased representation of phylum Firmicutes (Fig. 2a)[11,17]. Outgrowth of Proteobacteria has been associated with increased allo-BMT-related mortality due to GVHD[6,11,17], while abundance of genus *Blautia* was associated with reduced risk of death from GVHD[10]. Putting these data together with our aforementioned finding that allo-*Gpr43$^{-/-}$* animals showed worse GVHD mortality, we next set out to test the hypothesis that loss of GPR43 would steer the composition of the intestinal microbiota of *Gpr43$^{-/-}$* animals to a more inflammatory state when compared to the microbiota of WT B6 animals after allogeneic BMT. We characterized the microbiota of allo-WT B6 animals and the allo-*Gpr43$^{-/-}$* animals by *16S* rRNA gene sequence analysis and found a significant difference between the bacterial communities of the two groups at day +14 post allo-BMT ($p = 0.027$, Fig. 2a) with increased representation of the phylum Proteobacteria, and decreased representation of the class Clostridia (Fig. 2a), consistent with previous reports of their correlation with GVHD severity.

These findings raised the possibility that the effects of GPR43 expression in hosts on GVHD severity may be modulated by the alterations of the composition of the intestinal microbiota at the onset of BMT. To explore this possibility, we co-housed WT B6 and *Gpr43$^{-/-}$* animals for 28 days at 1:1 ratio prior to utilizing them as recipients in allo-BMT and determined fecal microbiota composition by 16S gene sequence analysis. The co-housed WT and *Gpr43$^{-/-}$* B6 animals showed similar fecal bacterial communities ($p = 0.199$, Fig. 2a, b). These data suggested that the microbial community observed in the *Gpr43$^{-/-}$* B6 animals were transferred to WT B6 animals by co-housing. After 28 days, the single or co-housed mice were used as recipients of syngeneic B6 or allogeneic MHC-mismatched allogeneic BALB/c donors as in Methods. The GPR43-deficient animals demonstrated greater GVHD than WT animals regardless of co-housing (Fig. 2c, d).

In separate experiments at a second institution (MSKCC), the WT and KO animals were co-housed for 7 days prior to BMT and the *16S* gene sequence analysis also revealed harmonization of the intestinal microbiome with no significant differences at day +1 post allo-BMT, and those communities were similar to that in naive *Gpr43$^{-/-}$* mice (Supplementary Fig. 1B). Furthermore, the fecal microbiota of co-housed GPR43$^{-/-}$ mice at day 14 after BMT also showed the increase of the phylum Proteobacteria and the loss of α diversity when compared to co-housed WT B6 mice (Supplementary Fig. 1B, C). Thus GPR43 expression in hosts

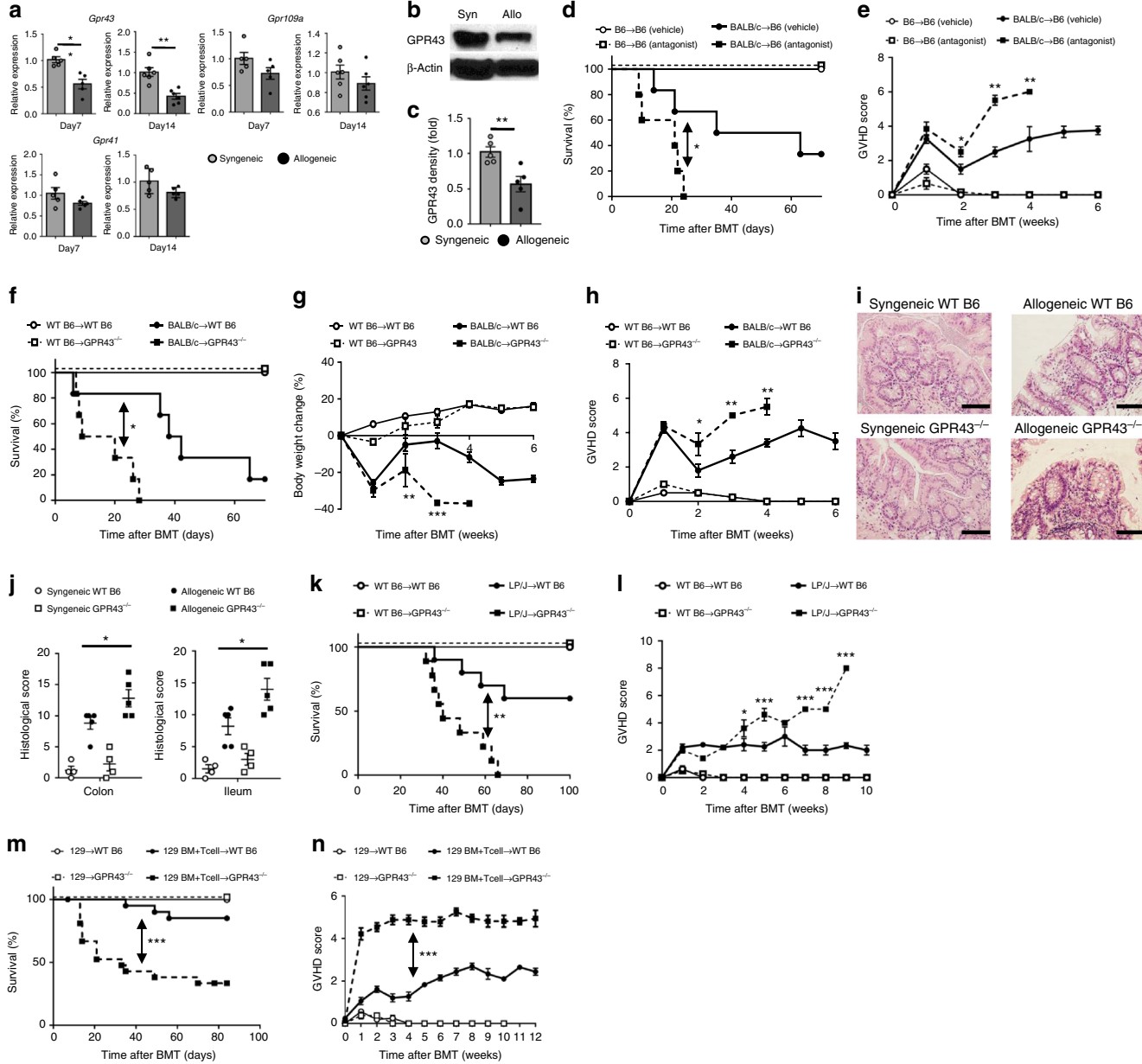

**Fig. 1** GPR43 protects against GVHD. B6 WT and *Gpr43−/−* mice received BMT from either syngeneic B6 or allogeneic BALB/c donors. **a** Gene expression of *GPR43*, *GPR41*, and *GPR109a* in IECs (CD326+) from syngeneic and allogeneic WT B6 recipients 7 and 14 days after BMT ($n = 5$ in Day 7, $n = 6$ in Day 14, two-tailed Mann–Whitney $U$ test). **b**, **c** Representative immunoblots and densitometric analysis of GPR43 normalized to the presence of β-actin in IECs (CD326+) from syngeneic and allogeneic WT B6 recipients 14 days after BMT ($n = 5$ each, two-tailed Mann–Whitney $U$ test). **d**, **e** Survival and clinical GVHD score after BMT treated with intragastric vehicle or GPR43 antagonists (10 mg kg−1 per day) from day 0 to day 21 ($n = 3$ each syngeneic, $n = 6$ allogeneic-vehicle, $n = 5$ allogeneic-antagonist, log-rank test for survival, two-tailed Mann–Whitney $U$ test for GVHD Score). **f–h** Survival, weight change, and clinical GVHD score after BMT ($n = 3$ each syngeneic, $n = 6$ each, log-rank test for survival, two-tailed unpaired *t*-test for body weight change and two-tailed Mann–Whitney $U$ test for GVHD Score). **i**, **j** Representative H&E stained colonic sections and histopathological GVHD score in colon and ileum from WT B6 or *Gpr43−/−* recipients on day 14 after BMT (scale bar, 100 μm) ($n = 4$ each syngeneic, $n = 5$ each allogeneic, two-tailed Mann–Whitney $U$ test). **k**, **l** WT B6 and *Gpr43−/−* mice received BMT from either syngeneic B6 or allogeneic LP/J donors. Survival and clinical GVHD score after BMT ($n = 5$ each syngeneic, $n = 10$ each allogeneic, log-rank test for survival, two-tailed Mann–Whitney $U$ test for GVHD Score). **m**, **n** WT B6 and *Gpr43−/−* mice received BMT from 129S1 donors. Survival and clinical GVHD score after BMT (129S1 BM to WT B6; $n = 4$, 129S1 BM to *Gpr43−/−*; $n = 3$, 129S1 BM + T cells to WT B6; $n = 21$, 129S1 BM + T cells to *Gpr43−/−*; $n = 15$, log-rank test for survival, two-tailed Mann–Whitney $U$ test for GVHD Score). Data are representative of three experiments. *$P < 0.05$, **$P < 0.01$, ***$P < 0.001$, Bars and error bars show the mean ± s.e.m.

ameliorates GVHD severity following experimental allo-BMT in a manner independent of the initial intestinal microbiota composition before BMT at two different institutions.

Antibiotic treatment is one of the main factors that changes the intestinal microbiota during the process of allo-BMT and has been shown to influence clinical GVHD[9,18]. We therefore

examined whether the alteration of the indigenous intestinal microbiota with antibiotic treatment had an impact on GVHD. The WT and *Gpr43−/−* B6 recipient animals were treated with an antibiotic cocktail from day 0 to until day 5 during allo-HCT as in Methods, and fecal microbiota was analyzed on days +7 and +14 post allo-HCT. Antibiotics caused significant differences between

the fecal bacterial communities (Fig. 2e). The fecal microbiota of antibiotic-treated animals in both groups showed decreased representation of butyrate producing Clostridiales species with an increased representation of the non-butyrogenic genus *Lactobacillus*, phylum Proteobacteria. These changes were more pronounced in the *Gpr43$^{-/-}$* animals than WT animals ($p = 0.022$ and $p = 0.033$ respectively, Fig. 2e). Importantly, antibiotic treatment led to significantly higher GVHD mortality in all of the allogeneic animals (Fig. 2f) confirming the relevance of

antibiotic induced microbial changes in GVHD severity in both WT and *Gpr43$^{-/-}$* recipients.

**GPR43 is critical for SCFA-mediated GVHD amelioration.** We next explored whether GPR43 signaling by an exogenous synthetic agonist enhanced survival from GVHD. To test this, we administered orally 10 mg kg$^{-1}$ per day GPR43 specific allosteric agonist to the recipients by gavage after BMT. There was no

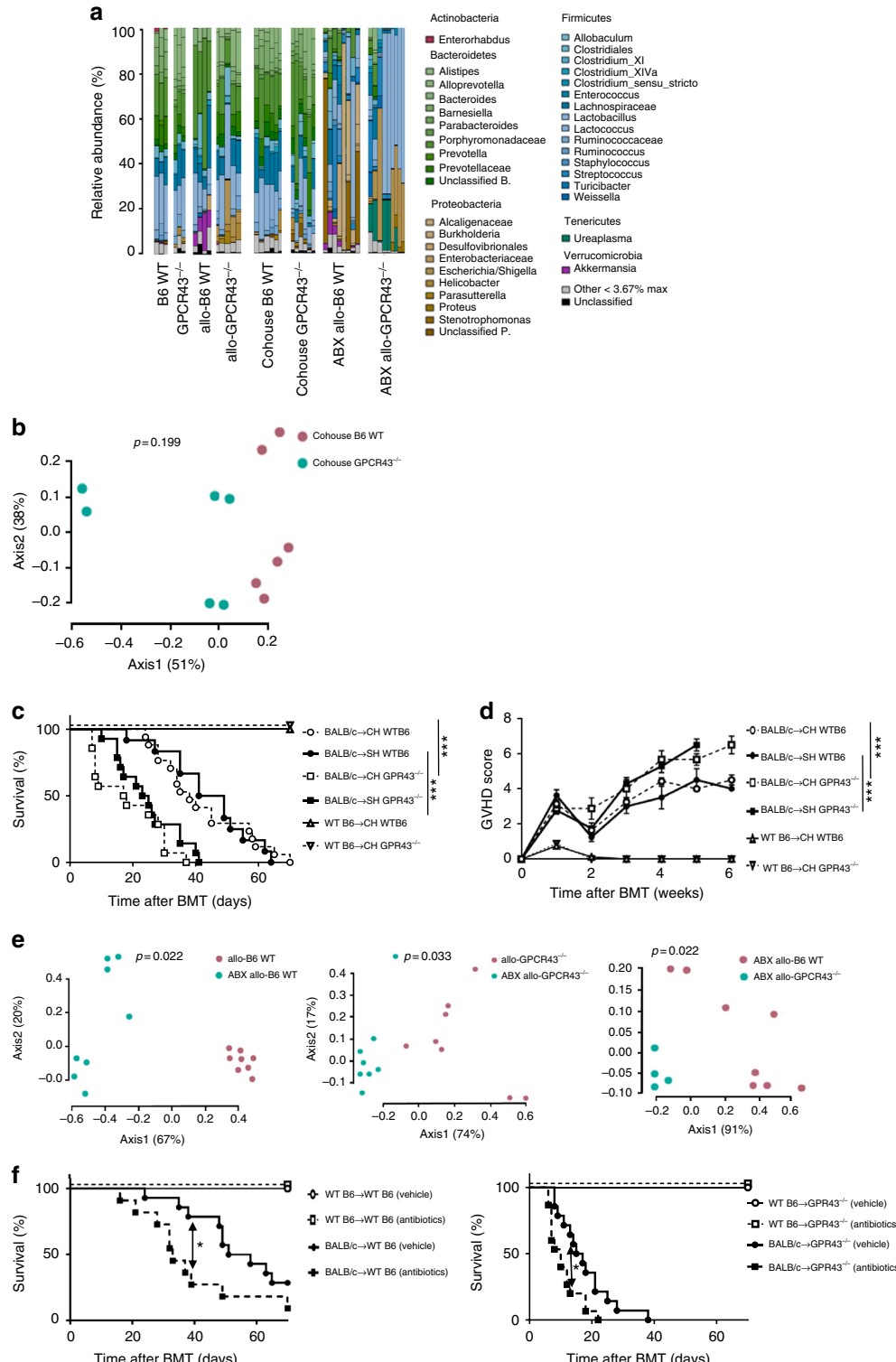

discernable toxicity because all syngeneic recipients survived. Oral gavage of GPR43 agonist (10 mg kg$^{-1}$ per day) significantly prolonged survival and reduced GVHD compared to vehicle-treated animals (Fig. 3a, b). By contrast, administration of the allosteric GPR43 agonist to allogeneic Gpr43$^{-/-}$ B6 animals did not mitigate GVHD (Fig. 3c, d), demonstrating the specificity of GPR43 in mitigating GVHD. To analyze the magnitude of improvement we next determined the effect of GPR43 agonist by enhancing the severity of GVHD in the same model system. Specifically, we increased GVHD severity by transplanting higher numbers of mature donor T cells (5 × 10$^6$ purified T cells) but kept all other transplant parameters similar to above. The recipient animals were then treated with the same dose (10 mg kg$^{-1}$ per day gavage) GPR43 agonist as above. The animals that were treated for similar duration as above, from days 0 to 7, showed modest, but statistically insignificant increase in survival when compared to control treated animals (Supplementary Fig. 2A-B). However, treatment over extended period of time, from days 0 to 21, demonstrated significantly greater survival than the control treated cohort (Supplementary Fig. 2A, B). Thus the magnitude of protection by the allosteric GPR43 agonist is dependent on the severity of GVHD.

We next tested whether administration of natural ligands of GPR43, namely the SCFAs, improved GVHD in a GPR43-dependent manner. To this end we orally administered SCFAs (acetate, butyrate, propionate) or the diluent by gavage to B6 allogeneic BMT recipients. Administration of butyrate reduced GVHD[13], as did propionate, although butyrate demonstrated overall better protection from GVHD (Fig. 3e–g). By contrast, administration of acetate, the most abundant SCFA in the intestinal lumen[1], had no effect on GVHD (Supplementary Fig. 2C-E) while administration of higher doses butyrate and propionate also provided no benefit to WT recipients (Supplementary Fig. 2F-G).

Administration of butyrate also reduced GVHD severity in Gpr43$^{-/-}$ B6 allogeneic animals, but its protection was less than in WT B6 animals (Fig. 3h, i). Protection from GVHD by butyrate is thus both GPR43 dependent and independent[13]. By contrast, administration of propionate to Gpr43$^{-/-}$ B6 animals failed to reduce GVHD severity, demonstrating its dependence on the presence of GPR43 (Fig. 3j, k). Collectively these data show that sensing of the microbial metabolites butyrate and propionate by host GPR43 plays a critical role in reducing GVHD.

**Cellular mechanisms of GPR43-dependent effects on GVHD.** We next explored the GPR43-dependent cellular mechanisms in regulation of GVHD. We examined in vivo markers of allogeneic T-cell activation and functions. Surprisingly, the total numbers of allogeneic donor T cells, IFN-γ$^+$ (Th1), TNF-α$^+$, IL-17A$^+$ (Th17) T cells, and regulatory T cell (Tregs) numbers from the spleen and intestinal tissues were similar between the WT and KO allogeneic recipients (Fig. 4a, b). The levels of serum inflammatory cytokines were also similar between allogeneic WT and Gpr43$^{-/-}$-recipient animals (Fig. 4c) demonstrating that absence of GPR43 in the hosts did not lead to systemic activation of donor T cells or inflammation.

Next, we sought to determine the critical GPR43 expressing cellular compartments for mediating its protective effects on GI GVHD severity. GPR43 is expressed in multiple cell types in the host, including immune cells such as the antigen-presenting cells (APCs) and non-immune cells such IECs, and also on donor T cells, all of which are critical in GVHD pathogenesis[19]. Host APCs such as macrophages and dendritic cells (DCs) isolated from colon and ileum of allogeneic WT recipients demonstrated greater expression of Gpr43 on day 7 after BMT (Supplementary Fig. 3A). Furthermore, in Gpr43$^{-/-}$ hosts, there was no compensatory increase in the expression of other SCFA receptors GPR41, GPR109a, or SLC5A8 in these cells when compared to those harvested from the WT hosts (Supplementary Fig. 3B), and did not show significant differences either in surface marker expression or in the ability to secrete TNF-α, IL-6, or IL-1β secretion following ex vivo stimulation by LPS (Fig. 4d, e). The ability of BMDCs and splenic DCs to stimulate allogeneic T cells was also similar between WT and Gpr43$^{-/-}$ groups when assessed in a mixed lymphoid reaction assay (Fig. 4f, Supplementary Fig. 3C). There were also similar numbers of macrophages and DCs in the spleen, liver, colon, and small intestine 7 days after transplantation in WT and Gpr43$^{-/-}$ animals (Fig. 4g). Furthermore, other antigen-presenting cells and neutrophils, specifically the myeloid cells, namely, CD11b$^+$ Ly6G$^+$ cells and CD11b$^+$ Ly6C$^+$ cells, were also similar in the WT and Gpr43$^{-/-}$ recipients at homeostasis and on days 7 and 14 after BMT in the spleen, colon, and small intestine (Supplementary Fig. 3D). These data suggest that the increase in GVHD severity in the absence of GPR43 expression in the hosts is unlikely to be due to measurable functional differences in host APCs.

We next assessed the expression and function of GPR43 on donor T cells. In contrast to host APCs, the allogeneic donor T cells demonstrated reduced expression of Gpr43 in vivo (Supplementary Fig. 3E) and expression of GPR43, GPR41, SLC5A8, and GPR109a in vitro (Fig. 4h). However, both the WT and the KO T cells demonstrated similar proliferation following anti-CD3/CD28 stimulation (Fig. 4i). Furthermore, both WT and Gpr43$^{-/-}$ Tregs suppressed WT effector T cells equally (Fig. 4j). Furthermore, there was no difference in GVHD severity when BMT hosts were transplanted with allogeneic T cells from either WT or Gpr43$^{-/-}$ B6 donors demonstrating that GPR43 expression on donor T cells does not contribute to GVHD severity (Fig. 4k, l).

**Fig. 2** Bacterial 16S rRNA-based analysis of the fecal microbiota of WT and Gpr43$^{-/-}$ mice. **a** Community composition bar plot of the fecal microbiota of naive WT B6 (B6 WT, n = 3) mice and naive Gpr43$^{-/-}$ mice (GPCR43$^{-/-}$, n = 3), WT B6 mice 14 days after allo-BMT (allo-B6 WT, n = 4), Gpr43$^{-/-}$ mice 14 days after allo-BMT (allo-GPCR43$^{-/-}$, n = 5), WT B6 mice 28 days after co-housing with Gpr43$^{-/-}$ mice (Cohouse B6 WT, n = 6), Gpr43$^{-/-}$ mice 28 days after co-housing with WT B6 mice (Cohouse GPCR43$^{-/-}$, n = 6), antibiotic-treated WT B6 mice 14 days after allo-BMT (ABX allo-B6 WT, n = 8) and antibiotic-treated Gpr43$^{-/-}$ mice 14 days after allo-BMT (ABX allo-GPCR43, n = 9). **b** Principle coordinates analysis (PCoA) of distances between communities of the fecal microbiota of six Cohouse B6 WT and six Cohouse GPCR43$^{-/-}$, P = 0.199 by analysis of molecular variance (AMOVA). **c, d** WT B6 and Gpr43$^{-/-}$ (single- or co-housed for 28 days before BMT) mice received BMT from either syngeneic allogeneic donors. Survival and clinical GVHD score after BMT (n = 3 each syngeneic, n = 12 allo SH WTB6, n = 17 allo CH WTB6, n = 14 allo CH Gpr43$^{-/-}$, n = 14 allo SH Gpr43$^{-/-}$, log-rank test for survival, two-tailed Mann–Whitney U test for GVHD Score, data are pooled from two experiments). **e** PCoA of distances between communities of the fecal microbiota of left; eight allo-B6 WT versus eight ABX allo-B6 WT, P = 0.022 by AMOVA, center; eight allo-GPCR43$^{-/-}$ versus eight ABX allo-GPCR43$^{-/-}$, P = 0.033 by AMOVA, right; four ABX allo-B6 WT versus eight ABX allo-GPCR43$^{-/-}$, P = 0.022 by AMOVA. **f** Survival after BMT for WT B6 (left) or Gpr43$^{-/-}$ (right) recipients treated with antibiotics (n = 3 each syngeneic, n = 12 each allogeneic WT B6, n = 14 each allogeneic Gpr43$^{-/-}$, log-rank test for survival). Data are representative of two experiments. *P < 0.05, ***P < 0.001, Bars and error bars show the mean ± s.e.m.

Because GPR43 expression on host APCs and donor conventional or regulatory T cells did not appear to be critical to reduce GVHD, we hypothesized that the expression of GPR43 in host non-hematopoietic cells, (such the GI tract where it is known to be highly expressed) is crucial for its ability to mitigate GVHD. We first analyzed whether the absence of GPR43 was associated

with any difference in the expression of other SCFA receptors in GVHD target organs such as colon, ileum, liver and skin, but did not find any differences between WT and $Gpr43^{-/-}$ animals (Supplementary Fig. 3F). Next, to evaluate the role of GPR43 expression in the host non-hematopoietic target tissues we generated [B6 $Gpr43^{-/-}$ → B6 WT] and [B6 WT →

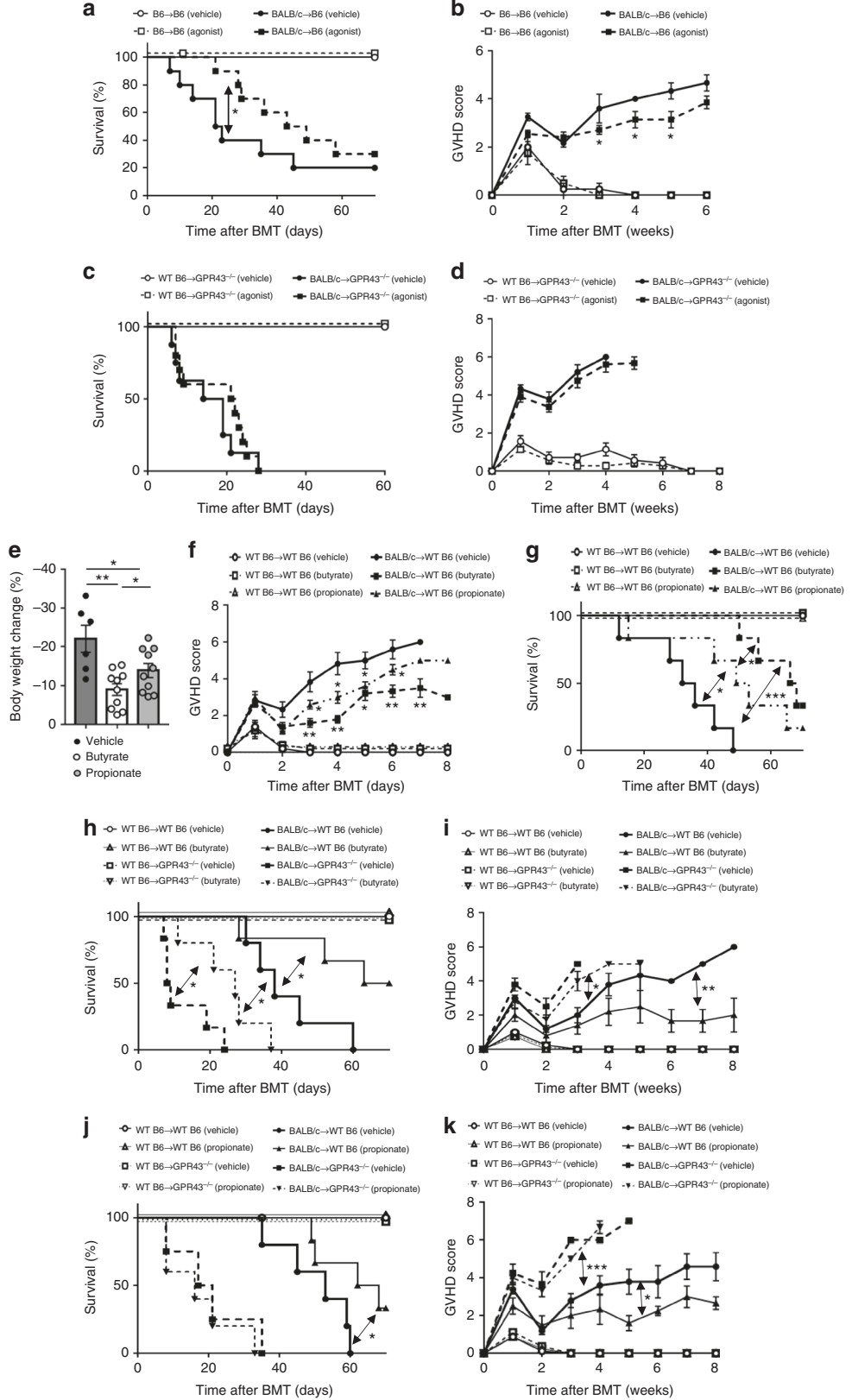

B6 Gpr43−/−] chimeric mice to limit GPR43 expression to either hematopoietic or non-hematopoietic cells. These chimeras [WTB6 → WT B6Ly5.2], [WT B6 Ly5.2 → Gpr43−/−], and [Gpr43−/− → WT B6Ly5.2] were then used 3 months later as BMT recipients (as in Methods). All chimeras that received syngeneic T cells and BM cells survived the duration of the observation period with no signs of GVHD (Fig. 4m, n). In contrast, the allogeneic [WT B6 → WT B6Ly5.2] and the [Gpr43−/− → WT B6Ly5.2] chimeras showed similar GVHD, while the [WT B6 Ly5.2 → Gpr43−/−] chimeras showed significantly more severe GVHD (P = 0.001).

Furthermore, the professional antigen-presenting cells subsets, namely macrophages and DCs, harvested from both [WT B6 → WT B6] and [WT B6 → Gpr43−/−] chimeras after irradiation were similar in absolute numbers, expression of surface marker, ability to secrete TNF-α, IL-6 following ex vivo stimulation by LPS (Supplementary Fig. 3G-I), and in their ability to stimulate allogeneic T cells (Supplementary Fig. 3H). Thus these data collectively demonstrate that expression of GPR43 on host non-hematopoietic cells is critical for controlling the severity of GVHD, although cannot formally rule out a contribution from immune cells.

We also analyzed serum LPS levels on day 14 after BMT as a marker for intestinal breach and found that LPS levels were significantly higher in Gpr43−/− recipients than in WT recipients following allo-BMT (Fig. 4o). The assessment of the intestinal permeability after allo-BMT was also determined by intragastric administration of fluorescein isothiocyanate (FITC)-dextran, a non-metabolized carbohydrate. Gpr43−/− allo-BMT recipients exhibited significantly high FITC-dextran in the serum at day 14 after allo-BMT (Fig. 4p).

**ERK phosphorylation depends on activation of NLRP3.** We next explored mechanisms downstream of GPR43 activation that might be relevant to GVHD. Sensing of the SCFA by the GPR43 activates the nucleotide-binding oligomerization domain (NOD)-like receptor protein 3 (NLRP3) inflammasome, which promotes IEC integrity and repair by increasing IL-18 secretion[20]. We thus determined whether GPR43 signaling altered cleaved IL-18 levels in IECs following allogeneic BMT. Both cleaved caspase-1 and IL-18 were reduced in CD326+ IECs harvested from Gpr43−/− animals when compared to those harvested from WT animals (Fig. 5a, b). Interestingly, GPR43 expression did not affect release of IL-1β, a cytokine considered redundant with respect to the protective effects of the inflammasome (Fig. 4e). To confirm NLRP3 inflammasome activation and IL-18 production is due to sensing of SCFAs by GPR43, we treated WT and Gpr43−/− mice with propionate following allo-BMT. Western blots of IECs

and serum cytokine levels on day +7 post allo-BMT showed increased levels of cleaved caspase-1 and IL-18 in the propionate-treated recipients (Fig. 5c, d). In contrast, IECs from Gpr43−/− mice showed reduced levels of cleaved caspase 1 and IL-18 regardless of treatment with propionate (Fig. 5c, d). These results were further validated by immunohistochemistry analysis for IL-18 in the IECs from propionate-treated WT mice on day 14 (Fig. 5e).

IL-22 secreted by innate lymphoid cells intestinal stem cell regeneration and protects from GI GVHD[21,22]. We therefore determined IL-22 levels in serum, colon, and ileum tissues at day 14 after BMT. Syngeneic WT and Gpr43−/− recipients in syngeneic groups demonstrated similar amounts of IL-22 (Fig. 5f) suggesting that the KO animals do not have a deficiency in IL-22 secretion in the absence of alloreaction. As demonstrated previously[21,22], WT allogeneic animals demonstrated a near complete absence of IL-22 (Fig. 5f). Similar reduction of IL-22 was also observed in Gpr43-deficient allogeneic animals (Fig. 5f). The IL-22 gene expression was also equivalently high in the lamina propria cells from both WT and Gpr43−/− syngeneic animals, and was decreased to similar levels in WT and Gpr43−/− allogeneic mice animals (Supplementary Fig. 4A). Thus within the limitations of the protein and mRNA assays, the reduction in the level of IL-22 was not significantly different in the allo-recipients of the WT and Gpr43−/−. However, whether IL-22 protects WT animals better than Gpr43−/− animals will need to be determined in future studies.

We next explored the upstream signaling mechanisms of GPR43-mediated activation of NLRP3. GPR43 couples to both Gαi and Gαq proteins[23]. Gαi inhibits adenyl cyclase pathway and cAMP production[20,24] while Gαq results in activation of extracellular signal-regulated kinase (ERK)[25,26]. However, whether SCFAs are sensed by Gαq of GPR43 on IECs and that in turn results in ERK dependent activation of NLRP3 is not known[27,28,29]. We therefore evaluated the role of GPR43-ERK dependent activation of NLRP3 in IECs by comparing SCFA-dependent phosphorylation of ERK in WT and GPR43−/− mice.

Treatment with propionate increased phospho-ERK levels in WT IECs but not in Gpr43−/− IECs (Fig. 5g, Supplementary Fig. 4B), suggesting that ERK activation is enhanced by SCFA-dependent GPR43 signaling. Next we analyzed whether IL-18 secretion was dependent on ERK phosphorylation-mediated activation and found IL-18 production to be suppressed by the ERK inhibitor in both propionate- and butyrate-treated IECs (Fig. 5h).

We next determined whether NLRP3 is the critical downstream pathway for mitigation of GVHD by the GPR43. NLRP3 expression was not changed at the protein level in Gpr43−/− recipient colon and ileum 14 days post allo-BMT relative to WT

**Fig. 3** GVHD-protective effect of SCFAs is mediated by GPR43. WT B6 and Gpr43−/− mice received BMT from either syngeneic B6 or allogeneic BALB/c donors. **a**, **b** Survival and clinical GVHD score after BMT for WT B6 recipients treated with vehicle or GPR43 agonists (10 mg kg−1 per day) from day 0 to day 21 (n = 5 each syngeneic, n = 10 each allogeneic, log-rank test for survival, two-tailed Mann–Whitney U test for GVHD Score). **c**, **d** Survival and clinical GVHD score after BMT for Gpr43−/− recipients treated with vehicle or GPR43 agonists (10 mg kg−1 per day) from day 0 to day 21 (n = 5 each syngeneic, n = 8 allogeneic-vehicle, n = 10 allogeneic-Agonist, log-rank test for survival, two-tailed Mann–Whitney U test for GVHD Score). **e–g** Body weight change on day 21 after BMT, survival and clinical GVHD score after BMT for WT B6 recipients treated with vehicle, butyrate or propionate from day 0 to day 21 (**e**, n = 6 vehicle, n = 9 butyrate, n = 9 propionate, **f**, **g**, n = 5 each syngeneic, n = 6 vehicle, n = 6 butyrate and propionate each, log-rank test for survival, two-tailed unpaired t-test for body weight change and two-tailed Mann–Whitney U test for GVHD Score). **h**, **i** Survival and clinical GVHD score after BMT for WT B6 and Gpr43−/− recipients treated with vehicle or butyrate (10 mg kg−1 per day) from day 0 to day 21 (n = 5 syngeneic each, n = 5 allogeneic WT-Vehicle and Gpr43−/−-Butyrate, n = 6 allogeneic WT-Butyrate and Gpr43−/−-Vehicle, log-rank test for survival, two-tailed Mann–Whitney U test for GVHD Score). **j**, **k** Survival and clinical GVHD score after BMT for WT B6 and Gpr43−/− recipients treated with vehicle or propionate (15 mg kg−1 per day) from day 0 to day 21 (n = 5 syngeneic each, allogeneic WT-Vehicle and Gpr43−/−-Propionate, n = 4 allogeneic Gpr43−/−-Vehicle, n = 6 allogeneic WT-, log-rank test for survival, two-tailed Mann–Whitney U test for GVHD Score). Data are representative of three experiments. *P < 0.05, **P < 0.01, ***P < 0.001, error bars show the mean ± s.e.m.

controls (Fig. 6a). To assess the influence of GPR43 on NLRP3 activation, we analyzed IL-18 production from WT or $Gpr43^{-/-}$ colon or ileum-derived IECs ex vivo. Monosodium urate (MSU), which directly stimulates NLRP3, was used as a positive control. IL-18 production was reduced in vehicle-treated $Gpr43^{-/-}$ IECs

relative to WT controls (Fig. 6b). This effect was specific to GPR43 as treatment of both WT and $Nlrp3^{-/-}$ IECs produced similar amounts of IL-18 with MSU (Fig. 6b) and the reduced IL-18 by $Gpr43^{-/-}$ IECs was thus not due to an intrinsic defect in the NLRP3 inflammasome. To determine whether

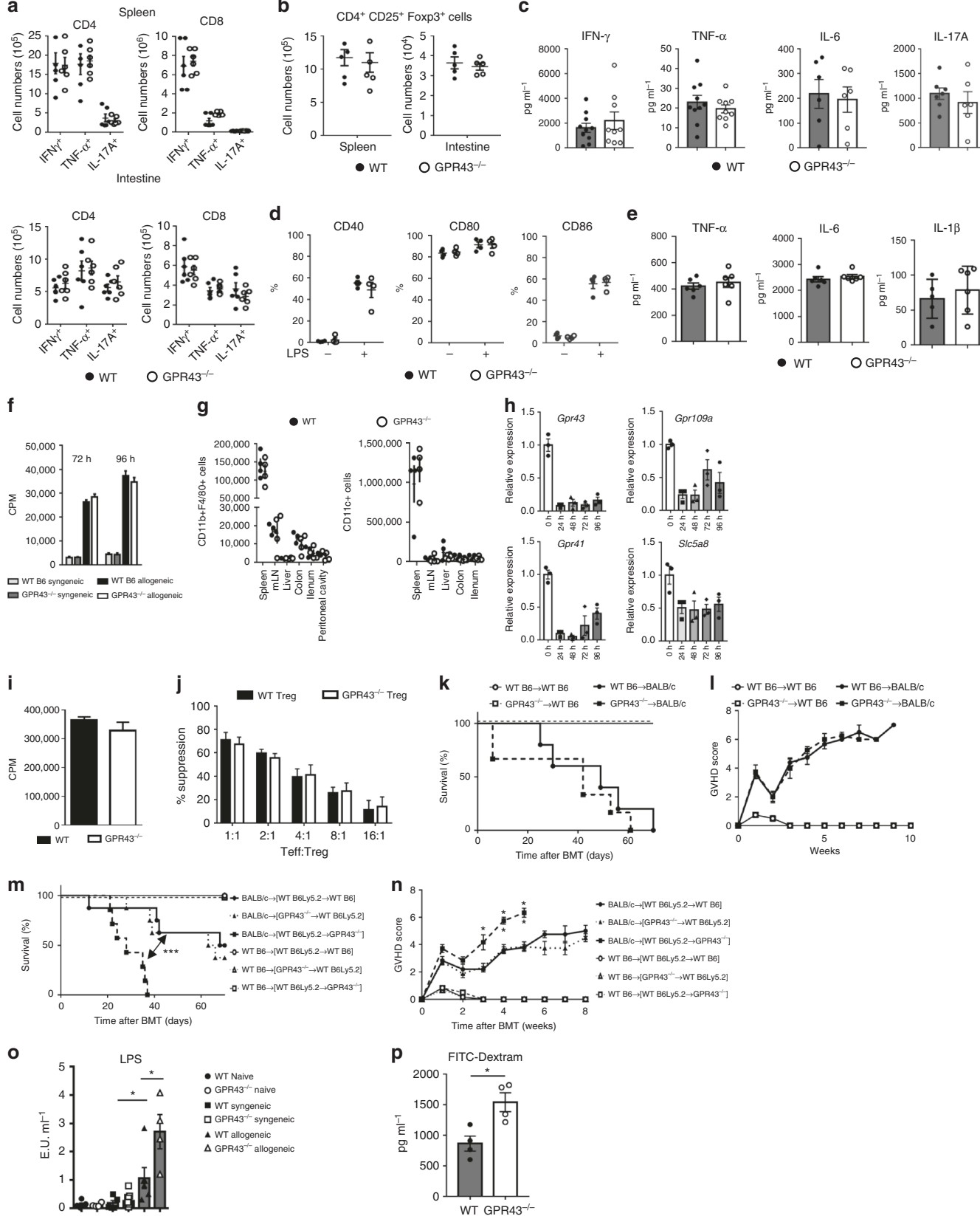

NLRP3 activation in non-hematopoietic cells was critical for resistance of GVHD by IECs, we generated chimeric mice which do not express NLRP3 only in the non-hematopoietic cells. These chimeric [WT B6 → WT B6Ly5.2] and [WT B6Ly5.2 → Nlrp3$^{-/-}$] animals were used 3 months later as recipients of allogeneic BMT. All chimeras that received syngeneic T cells and BM cells survived with no signs of GVHD. The allogeneic [WT B6Ly5.2 → Nlrp3$^{-/-}$] chimeras that lacked NLRP3 on host non-hematopoietic target tissues demonstrated significantly greater severity of GVHD (P = 0.001) compared with [WT B6 → WT B6Ly5.2] recipients (Fig. 6c, d). To verify the influence of GPR43 dependent activation of the NLRP3 inflammasome for GVHD resistance, we administered propionate, which requires GPR43 to reduce GVHD (Fig. 3h, i), or butyrate, which only partially requires GPR43 for its anti-GVHD activity (Fig. 3f, g), to either [WT B6 → WT B6Ly5.2] or [WTB6 Ly5.2 → Nlrp3$^{-/-}$] recipients following allo-BMT. Similar to the Gpr43$^{-/-}$ results shown in Fig. 3f, h, propionate treatment resulted in no change in GVHD severity whereas butyrate treatment partially ameliorated GVHD in [WT B6 → Nlrp3$^{-/-}$] recipients (Fig. 6e, f). Collectively, these data indicated that SCFA-mediated activation of NLRP3 inflammasome in host non-hematopoietic cells is critical for reducing the severity of GVHD (Fig. 7).

## Discussion

The biology underpinning the microbiome alterations and the severity of GVHD remain poorly understood. Correction of the decrease in certain microbiome derived metabolites such as the SCFA butyrate following allo-BMT reduces GVHD[1,13]. However, the mechanisms of sensing of butyrate and other microbial SCFAs that are critical for reduction of GVHD remain poorly understood. We demonstrate utilizing combination of genetic, chemical loss and gain of function approaches that GPR43 is the critical sensor of SCFAs butyrate and propionate and demonstrate a novel role for SCFA–GPR43–ERK–NLRP3 axis in mitigating GVHD (summarized in Fig. 7).

Our data underscores the importance of SCFAs for gut protection from GVHD, and outlines a precise SCFA receptor (GPR43) signaling pathway involving NLRP3 inflammasome activation in recipient non-hematopoietic cells (presumably IECs) that results in resistance to GVHD. Expression of the SCFA receptor GPR43 is decreased in IECs early after allo-BMT, and Gpr43$^{-/-}$ mice were more susceptible to GVHD. Loss of GPR43 led to severe tissue damage without changes of infiltrating T-cells or inflammatory cytokines. The importance of GPR43 for GVHD protection was further confirmed by administration of a synthetic antagonist and agonist to GPR43. In contrast to previous reports utilizing cell lines, our data show that the synthetic GPR43 antagonist is an effective blocker of GPR43 on IECs in vitro and in GVHD vivo (Supplementary Fig. 5)[30].

The effect of GPR43 deficiency on GVHD was independent of the initial microbial architecture as homogenization of intestinal microbiota after co-housing WT and Gpr43$^{-/-}$ mice failed to improve GVHD in the Gpr43$^{-/-}$ animals. However experiments abrogating SCFA-producing bacteria by antibiotic administration and those by direct administration of SCFA butyrate and propionate, demonstrate that SCFAs protect against GVHD in part through their sensing by the GPR43 receptor. It is important to note that the data presented here do not exclude a role for other SCFA receptors or transporters such as GPR41 and GPR109a despite the lack of alteration in their expression after allo-HCT. The further augmentation of GVHD observed in Gpr43$^{-/-}$ animals upon depletion of SCFA-producing bacteria with antibiotics might suggest a role for detection of butyrate by other GPR proteins.

Much remains to be understood about microbiome, its changes, diversity and the implications for GI GVHD. Specifically, the cause for GI dysbiosis and loss of diversity after allo-BMT remains unknown. Furthermore, whether dysbiosis after BMT is a cause, a consequence, or is an amplifier of GVHD remains unclear. The presence of various microbiota-derived metabolites, and their relationship to host diet remain unknown. Our data add some clarity and nuance to the current understanding of the role of gut dysbiosis while also raising more questions. They show that metabolites generated by the GI microbiome, specifically SCFAs, and not just butyrate, but also propionate, have salutary effects on GI GVHD. They also demonstrate that these metabolites will need to be sensed by appropriate receptors, namely GPR43. SCFAs are also sensed by GPR proteins, such as GPR109a etc. Thus the collective effects of SCFAs butyrate and propionate is likely the net result of their sensing by several GPR proteins. Our data show that except for GPR43, the expression of other SCFA sensing GPRs was not altered after allo-BMT. Because GPR43 deficiency alone aggravates GVHD while its specific agonist alleviates GVHD (the agonist, unlike SCFAs, would not work via other GPRs) suggest that protection via GPR43 is pivotal for GI GVHD. GPR43 can heterodimerize with GPR41 for signaling, but our data show that only GPR43 expression was reduced after allo-BMT suggesting that regulation of GPR43, transcriptionally and/or translationally, after allo-BMT might be unique when compared to GPR41 or other GPRs. The reasons for such specific reduction and regulation of GPR43 after allo-BMT will need to be determined in future studies. Furthermore, whether absence of any other GPR proteins, such as GPR109a or GPR41, in the presence of sufficient amount of GPR43, can also aggravate

**Fig. 4** Expression of GPR43 in non-hematopoietic cells is necessary for GVHD-protective effect. WT B6 and Gpr43$^{-/-}$ mice received BMT. **a** Total number of donor IFN-γ, TNF-α, and IL-17A positive T cells (n = 6 each). **b** Total numbers of CD4$^+$CD25$^+$Foxp3$^+$ Treg cells, (n = 5 each, two-tailed unpaired t-test). **c** Serum IFN-γ, TNF-α, IL-6, and IL-17A levels from allogeneic recipients on day 14 after BMT (n = 6 each). **d** BMDCs treated with LPS were analyzed for CD40, CD80, and CD86 populations (n = 4 each, two-tailed unpaired t-test). **e** TNFα, IL-6, and IL-1β production by BMDCs stimulated overnight with LPS (n = 6 each, two-tailed unpaired t-test). **f** T cells from WT B6 or BALB/c were cultured with irradiated BMDCs for 72 and 96 h and analyzed for proliferation (n = 5 each, two-tailed unpaired t-test). **g** Total number of CD11b$^+$ F4/80$^+$ and CD11c$^+$ 14 days after BMT (n = 4 each, two-tailed unpaired t test). **h** T cells from WT B6 incubated with anti-CD3 (2 μg per ml) and anti-CD28 (1 μg per ml) antibodies. Gene expression in 24, 48, 72, and 96 h after stimulation (n = 3 each). **i** Proliferation of stimulated splenic T cells for 72 h (n = 5 each, two-tailed unpaired t-test). **j** Irradiated splenocytes from BALB/c mice were co-cultured with effector T cells and Treg cells at different ratios and analyzed for T-cell proliferation (n = 3 each). **k, l** WT BALB/c mice received BMT. Survival and clinical GVHD score after BMT (n = 5 syngeneic and WT allogeneic each, n = 6 Gpr43$^{-/-}$ allogeneic, log-rank test for survival, two-tailed Mann–Whitney U test for GVHD Score). **m, n** Chimeric [B6 → B6 Ly5.2], [Gpr4343$^{-/-}$ → B6 Ly5.2], and [B6 Ly5.2 → Gpr43$^{-/-}$] animals received BMT. Survival and clinical GVHD score after BMT (n = 3 syngeneic each, n = 8 allogeneic each, log-rank test for survival, two-tailed Mann–Whitney U test for GVHD Score). **o** Serum LPS levels at day 14 after BMT (n = 4 naive, allogeneic, WT syngeneic, n = 10 Gpr43$^{-/-}$ syngeneic, two-tailed unpaired t-test). **p** Serum FITC-dextran levels at day 14 after BMT (n = 4 each, two-tailed unpaired t-test). *P < 0.05, **P < 0.01, ***P < 0.001, error bars show the mean ± s.e.m. Data are representative of two or three experiments

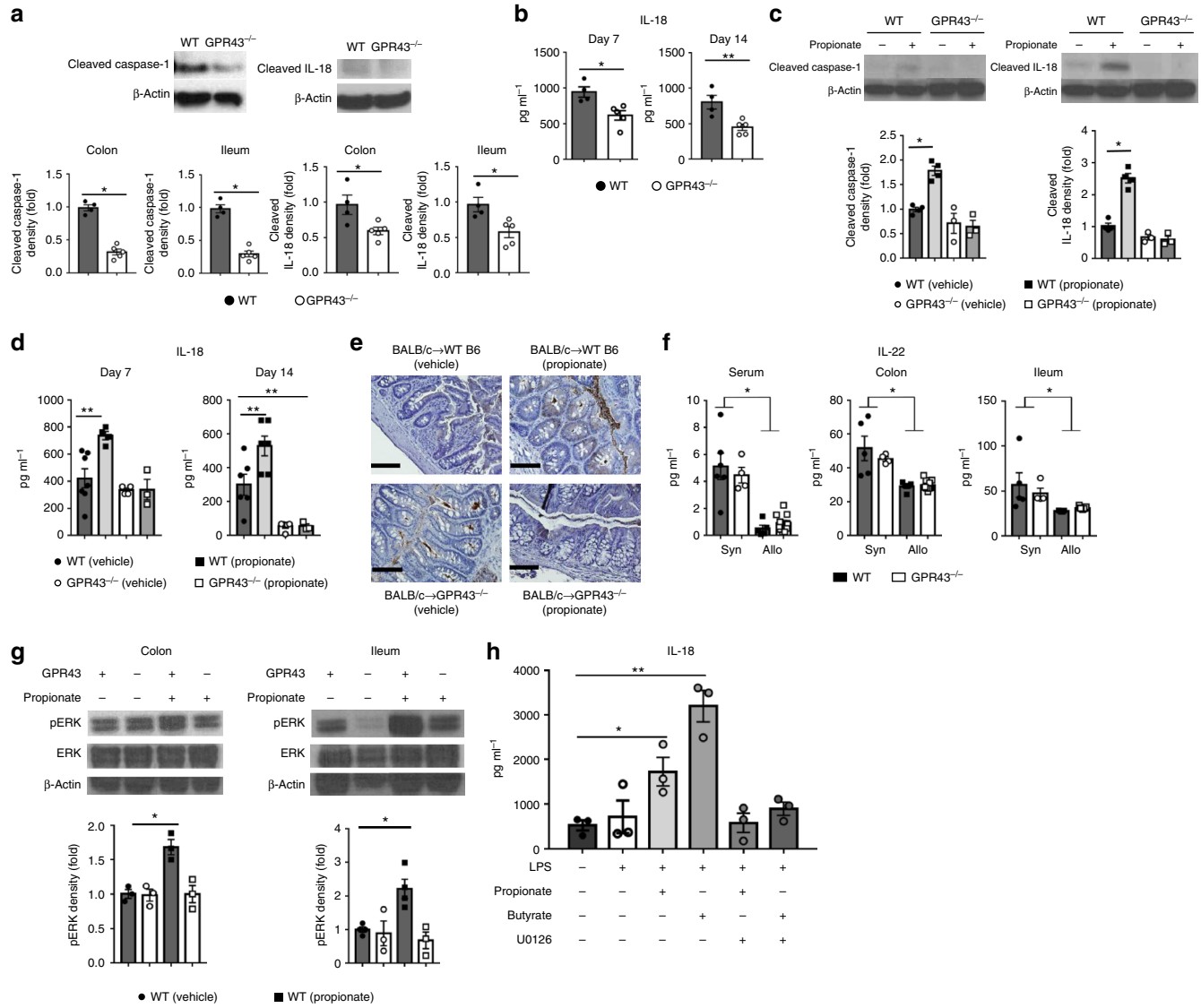

**Fig. 5** SCFA-dependent GPR43 signaling confers GI GVHD protection through the ERK–NLRP3 pathway. B6 WT and $Gpr43^{-/-}$ mice received BMT from syngeneic or allogeneic donors. **a** Representative immunoblots and densitometric analysis of cleaved caspase 1 and cleaved IL-18 normalized to the presence of β-actin in IECs (CD326$^+$) from allogeneic recipients 14 days after BMT ($n = 4$–5 each, two-tailed Mann–Whitney U test). **b** Serum IL-18 levels at day 7 and 14 after BMT ($n = 4$ WT, $n = 5$ $Gpr43^{-/-}$, two-tailed unpaired $t$-test). **c** Representative immunoblots and densitometric analysis of cleaved caspase 1 and cleaved IL-18 at day 14 after BMT in IECs (CD326$^+$) from allogeneic recipients. Mice were treated with vehicle or propionate (15 mg kg$^{-1}$ per day) from day 0 ($n = 4$ WT, $n = 3$ $Gpr43^{-/-}$, two-tailed Mann–Whitney U test). **d** Serum IL-18 levels from allogeneic recipients treated with vehicle or propionate (15 mg kg$^{-1}$ per day) at day 7 and 14 after BMT ($n = 5$ WT each, $n = 4$ $Gpr43^{-/-}$ each, two-tailed unpaired $t$-test). **e** Immunohistochemical staining of IL-18 in colon sections from WT B6 and $Gpr43^{-/-}$ recipients 14 days after BMT treated with vehicle or propionate (15 mg kg$^{-1}$ per day) from day 0 ($n = 3$–5 each). **f** IL-22 levels in serum, homogenized tissues from syngeneic and allogeneic recipients at day 14 after BMT ($n = 6$ WT syn, $n = 4$ $Gpr43^{-/-}$ syn, $n = 5$ WT allo, $n = 10$ $Gpr43^{-/-}$ allo, two-tailed unpaired $t$-test). **g** WT B6 and $Gpr43^{-/-}$ mice received oral administrations of vehicle or propionate (15 mg kg$^{-1}$ per day) for 3 days. Representative immunoblots and densitometric analysis of phosphorylated ERK normalized to total ERK in IECs (CD326$^+$) after treatment ($n = 4$ WT each, $n = 3$ $Gpr43^{-/-}$ each, two-tailed Mann–Whitney U test). **h** Colon explants from WT B6 mice were pretreated with U0126 (10 µM) for 30 min. Then tissues were primed with LPS (0.1 µg per ml) for 2 h and activated with butyrate or propionate (1 mM) for 2 h. Supernatants were assayed for IL-18 by ELISA ($n = 3$ each, two-tailed unpaired $t$-test). Data are representative of two to three experiments. *$P < 0.05$, **$P < 0.01$, ***$P < 0.001$, error bars show the mean ± s.e.m.

GVHD remains to be determined. Our data raise the possibility that host diets, which lead to generation of appropriate SCFAs, such as those rich is fiber, might play a role in mitigating GVHD. Interestingly, unlike propionate and butyrate, we did not see improvement in GVHD when the other SCFA, acetate, was administered to the recipients. While speculations such as that acetate levels may be optimal and sufficiently available are plausible, the reason for these results cannot be fully explained and

will need to be carefully explored in future studies. It is also important to note that while our study demonstrates beneficial effects of microbiome derived metabolites, the SCFAs, the impact of all other microbiome derived metabolites, whether they are salutary or harmful after allo-BMT, will remain to be methodically analyzed in future studies.

Amongst the SCFAs, GPR43 is important for the anti-GVHD effects of propionate and butyrate but not acetate. In addition,

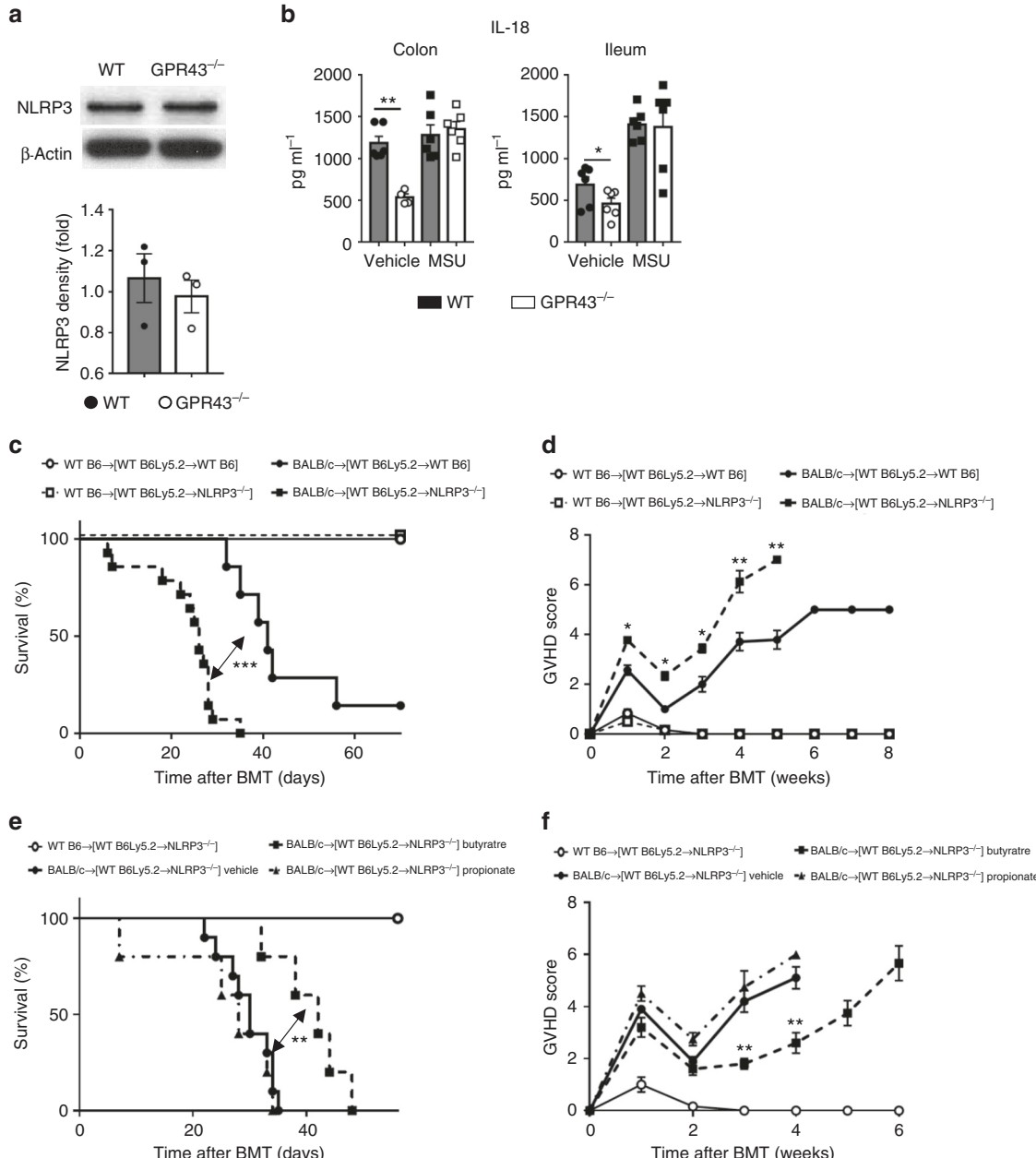

**Fig. 6** NLRP3 deficiency in non-hematopoietic cells exacerbates GVHD and is not rescued by propionate. B6 WT and *Gpr43*−/− mice received BMT from either syngeneic B6 or allogeneic BALB/c donors. **a** Representative immunoblots and densitometric analysis of NLRP3 normalized to the presence of β-actin in IECs (CD326+) from allogeneic recipients 14 days after BMT (*n* = 3 each, two-tailed Mann–Whitney *U* test). **b** IL-18 production by colon and ileum explant culture from WT B6 and *Gpr43*−/− mice stimulated overnight with MSU as measured by ELISA (*n* = 6 each, two-tailed unpaired *t*-test). **c**, **d** Chimeric [B6Ly5.2 → B6] and [B6 Ly5.2 → *Nlrp3*−/−] animals received BMT from either syngeneic WT B6 or allogeneic BALB/c donors. Survival and clinical GVHD score after BMT (*n* = 6 syngeneic each, *n* = 7 allogeneic [B6Ly5.2 → B6], *n* = 14 [B6 Ly5.2 → *Nlrp3*−/−], log-rank test for survival, two-tailed Mann–Whitney *U* test for GVHD Score) are depicted. Data are pooled from two experiments. **e**, **f** Chimeric [B6Ly5.2 → B6] and [B6 Ly5.2 → *Nlrp3*−/−] animals after BMT were treated with vehicle, butyrate (10 mg kg−1 per day) or propionate (15 mg kg−1 per day). Survival and clinical GVHD score after BMT (*n* = 6 syngeneic, *n* = 10 Vehicle, *n* = 5 Butyrate and Propionate each, log-rank test for survival, two-tailed Mann–Whitney *U* test for GVHD Score). Data are pooled from two experiments. **P* < 0.05, ***P* < 0.01, ****P* < 0.001, error bars show the mean ± s.e.m.

butyrate appeared to have both GPR43 dependent and independent anti-GVHD mechanisms, whereas propionate's anti-GVHD effect required GPR43. The likely GPR43-independent anti-GVHD mechanism of butyrate includes HDAC inhibition and acting as a nutritional source for IECs[13]. The lack of impact of acetate suggests it may have additional distinct effects on IECs or the lack of benefit from replenishing the already sufficient amount of acetate[1,13].

We failed to find an impact of GPR43 deficiency on APC (DC/Macrophages), effector T cell, or regulatory T-cell function after allo-HCT. However, using chimeric mice, we observed that GPR43 expression on non-hematopoietic cells, presumably IECs because of significant reduction in GI GVHD. Previous work has shown that optimal GPR43 stimulation causes activation of the NLRP3 inflammasome in gut epithelium, which contributes to gut homeostasis by generating the IEC-protective cytokine,

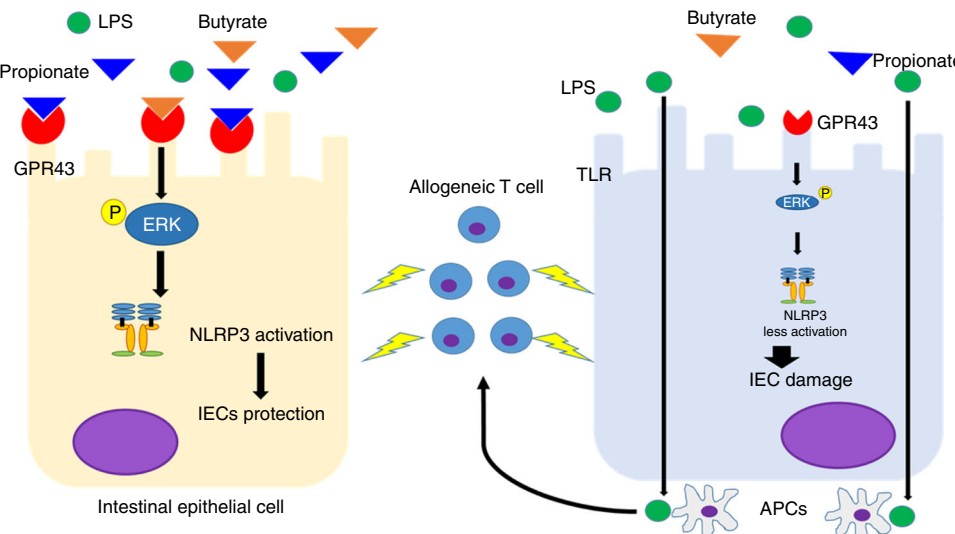

**Fig. 7** Model illustrating role of GPR43 on IECs after allo-BMT. In the context of injury and inflammation caused by alloreactive T cells and PAMPs such as LPS, SCFAs engage GPR43 and promote ERK phosphorylation dependent activation of NLRP3 in IECs that protects them and mitigates damage induced by GVHD (left). Reduction in expression of GPR43 reduces ERK phosphorylation dependent activation of NLRP3 and aggravates GVHD induced IEC damage and barrier breach allowing for greater translocation of PAMPs such as LPS which further amplify the inflammation and damage (right)

IL-18[21]. Here we build on this prior mechanistic work and demonstrate that GPR43 signaling following binding of propionate or butyrate also activates the NLRP3 inflammasome leading to greater IL-18 production after allo-BMT. Importantly, intrinsic NLRP3 function remained intact in our experimental system as direct stimulation of NLRP3 with MSU, that bypasses GPR43, was not reduced in $Gpr43^{-/-}$ IECs. We demonstrate that GPR43 signaling, potentially through the Gαq subunit, promotes ERK phosphorylation, which is critical for activation of NLRP3 inflammasome activation. Finally, we demonstrate that recipient NLRP3 expression in non-hematopoietic tissues is critical for propionate-mediated resistance of GVHD. The role of NLRP3 inflammasome in host hematopoietic antigen-presenting cells has previously been demonstrated to play a role in enhancing GVHD[31]. Our data complement and add texture to the role of NLRP3 in GVHD by demonstrating that its activation in non-hematopoietic cells has converse effects and point to the complex nature of inflammasome in regulation or aggravation of GVHD.

In summary, our data provide mechanistic insights into microbial metabolite-mediated regulation of GVHD and identify a critical role for GPR43–ERK–NLPR3 axis in non-hematopoietic host cells such as IECs (Fig. 7) in mitigating severity of target tissue damage.

## Methods

**Mice**. C57BL/6 (B6, H-2[b], CD45.2[+]), B6 Ly5.2 (H-2[b], CD45.1[+]), and LP/J (H-2[b]) mice were purchased from the Jackson Laboratory (Bar Harbor, ME, USA). BALB/c (H-2[d]) mice were purchased from Charles River Laboratories (Wilmington, MA, USA). B6-background $GPR43^{+/-}$ mice were purchased from Deltagen (San Mateo, CA, USA) and bred to obtain $GPR43^{-/-}$ mice. $Nlrp3^{-/-}$ were bred in the University of Michigan[32]. Female mice were used for experiments during 8–12 weeks old. All mice were kept under specific pathogen-free (SPF) conditions at the University of Michigan Cancer Center. All animals were cared for according to regulations reviewed and approved by the University of Michigan Committee on the Use and Care of Animals, which are based on the University of Michigan Laboratory Animal Medicine guidelines.

**Generation of bone marrow (BM) chimeras**. WT B6, B6Ly5.2, $Gpr43^{-/-}$, and $Nlrp3^{-/-}$ animals were subjected to 1000 cGy total-body irradiation (TBI) from a [137]Cs source on day −1 and then injected intravenously with $5 \times 10^6$ BM cells from WT B6, B6 Ly5.2, or $Gpr43^{-/-}$ donor mice on day 0 [33,34]. Donor hematopoietic chimerism was confirmed using a CD45.2 monoclonal antibody 3 months after BMT.

**Bone marrow transplantation**. Splenic T cells from donors were enriched, and the BM was depleted of T cells by autoMACS (Miltenyi Biotec, Bergisch Gladbach, Germany) utilizing CD90.2 microbeads (Miltenyi Biotec)[33–36]. WT C57BL/6, $Gpr43^{-/-}$, and WT BALB/c female animals were used as recipients and received either 800 cGy (BALB/c) or 1000 cGy (C57BL/6 and $Gpr43^{-/-}$) TBI on day −1, respectively, and $1 \times 10^6$ (C57BL/B6 or $Gpr43^{-/-} \to$ BALB/c), $4 \times 10^6$ (LP/J $\to$ C57BL/6 or $Gpr43^{-/-}$), or $2.5$–$5.0 \times 10^6$ (BALB/c $\to$ C57BL/6, $Gpr43^{-/-}$ or $Nlrp3^{-/-}$) CD90.2[+] T cells along with $5 \times 10^6$ T-cell-depleted BM (TCD-BM) cells from either syngeneic or allogeneic donors on day 0. [B6 → B6 Ly5.2], [$Gpr43^{-/-}$ → B6 Ly5.2], [B6 Ly5.2 → $Gpr43^{-/-}$], [B6Ly5.2 → B6], and [B6 Ly5.2 → $Nlrp3^{-/-}$] animals received 900 cGy TBI on day −1 and were injected intravenously with $2.5 \times 10^6$ CD90.2[+] T cells and $5 \times 10^6$ TCD-BM from either syngeneic B6 or BALB/c donors on day 0. Animals received vehicle or GPR43 antagonist (GLPG0974, 10 mg kg[−1] per day, Tocris, Minneapolis, MN, USA) or allosteric GPR43 agonist (10 mg kg[−1] per day, Millipore, San Diego, CA, USA) according to manufacturer's instructions by flexible 20-gauge, 1.5-in. intra-gastric gavage needle daily from day 0 to day 21[15,37]. Antibiotic treatment consisted of 5 days of 200 μl/mouse oral gavage of a cocktail containing ampicillin (1 mg per ml), gentamicin (1 mg per ml), metronidazole (1 mg per ml), kanamycin (1 mg per ml), and vancomycin (0.5 mg per ml) (Sigma-Aldrich)[38]. Animals received vehicle or the indicated doses of sodium acetate, butyrate or propionate (Sigma, St. Louis, MO) by flexible 20-gauge, 1.5-in. intra-gastric gavage needle daily from day 0 to day 21[13]. For BMTs performed at Memorial Sloan Kettering Cancer Center, WT C57BL/6J and $Gpr43^{-/-}$ mice were irradiated (11 Gy, split dose) and transplanted with 129S1 bone marrow cells ($5 \times 10^6$) and T cells ($1 \times 10^6$ CD5 MACS), and their condition was monitored for development of clinical GVHD. The mice were randomly assigned to treatment groups in each experiment. No mice were excluded from analysis. No statistical methods were used to predetermine sample size. The investigators were not blinded to allocation during experiments and outcome assessment.

**Systemic and histopathological analysis of GVHD**. We monitored survival after allo-HCT daily and assessed the degree of clinical GVHD weekly, as described previously[39]. Histopathological analysis of the liver, GI tract, and lung, which are the primary GVHD target organs, was performed as described utilizing a semiquantitative scoring system implemented in a blinded manner by a single pathologist (C.L.)[40].

**Cell isolation**. Primary IECs were obtained from C57BL/6J and $Gpr43^{-/-}$ mice after digestion once with 0.1 mM EDTA at 37°C for 45 minutes[13]. For splenic DC isolation, spleens were cut into thirds, flushed with 1 mg/mL collagense D (Roche, Germany) and incubated for 1 h at 37 °C[41]. Digested spleens were then homogenized between frosted slides, filtered through a 40 um cell strainer to achieve a single-cell suspension, and enriched with CD11c UltraPure MicroBeads (Miltenyi Biotec). T cells were isolated from spleens using CD90.2 MicroBeads (Miltenyi Biotec). For intestinal lymphocytes, DCs and macrophages isolation, intestinal tissues were treated with HBSS containing 1 mM dithiothreitol and 20 mM EDTA at 37 °C for 20 min. The tissues were then minced and dissociated with collagenase D and DNase I (Roche), at 37 °C for 30 min to obtain single-cell suspensions[42].

After filtering, the single-cell suspensions were subjected to Percoll gradient separation. Pan T cell Isolation Kit II (Miltenyi Biotec) for T cells, APC anti-F4/80 antibodies (Biolegend, BM8) and anti-allophycocyanin magnetic microbeads (Miltenyi Biotec Ltd) for macrophages, and CD11c UltraPure MicroBeads (Miltenyi Biotec) for DCs via autoMACS (Miltenyi Biotec) were used. Macrophages from the peritoneal cavity were obtained by washing with 5 ml of PBS containing 5 mM EDTA.

**qRT-PCR.** Total RNA from single-cell suspensions or snap-frozen skin, liver, colon or ileum was isolated using the miRNeasy Kit (Qiagen) and reverse transcribed into cDNA using the High Capacity cDNA Reverse Transcription Kit (Applied Biosystems, Foster City, CA). The following primers and PowerUP SYBR green polymerase (Applied Biosystems) were used to detect the following transcripts: 5′-CACGGCCTACATCCTCATCT-3′ and 5′-TTGGTAGGTACCAGCGGAAG-3′ (Gpr43); 5′-CTGGCGGAGCTACGTGCT-3′ and 5′-GGGGTCGATACAAGAGT-3′ (Gpr41)[43]; 5′-ATGGCGAGGCATATCTGTGTAGCA-3′ and 5′-TCCTGCCTGAGCAGAACAAGATGA-3′ (Gpr109a)[44]; 5′-GTCAACCG-CACCTTTATGCT-3′ and 5′-GAACAGTTTCTCCCCGATGA-3′ (IL-22); and 5′-TGACCTCAACTACATGGTCTACA-3′ and 5′-CTTCCCATTCTCGGCCTTG-3′ (Gapdh)[45]. All reactions were performed according to manufacturer's instructions. All primers were verified for the production of a single specific PCR product via melting curve analysis.

**Western blot analysis.** Whole-cell lysates were obtained and protein concentrations determined by a BCA protein assay (Thermo Fisher Scientific, Rockfold, IL). Protein was separated by SDS-PAGE gel electrophoresis and subsequently transferred to a PVDF membrane (EMD Millipore) using a Bio-Rad semi-dry transfer cell (20 V, 1 h). Blots were incubated with anti-GPR43 (Millipore, ABC299, 1.0 µg/ml), NLRP3 (Novus Biologicals, NBP2-12446, 5.0 µg/ml), Caspase-1 (Abcam, ab179515, 1:1000 dilution), IL-18 (MBL, D046-3, 1.0 µg/ml), phospho-p44/42 MAPK (Erk1/2) (Thr202/Tyr204, Cell Signaling Technology, 9101, 1:1000 dilution), p44/42 MAPK (Erk1/2) (137F5, Cell Signaling Technology, 4695, 1:2000 dilution) or β-actin (Abcam, ab8226, 1:3000 dilution) primary antibodies overnight at 4 ℃. Incubation with secondary anti-rabbit-HRP (Santa Cruz, sc-2357), anti-mouse-HRP (Santa Cruz, sc-2005), or anti-rat-HRP (Abcam, ab97057) was performed at room temperature for 1 h. Bound antibody was detected using Super-Signal ECL substrate (Thermo Fisher Scientific). Densitometric analysis was performed using ImageJ.

**Fluorescence-activated cell sorting analyses.** Fluorescence-activated cell sorting (FACS) analyses was performed as following; cells were re-suspended in FACS wash buffer (2% bovine serum albumin (BSA) in phosphate buffered saline (PBS) and stained with conjugated monoclonal antibodies[34,46]. The following antibodies were obtained from BioLegend: PerCP/Cy5.5-anti-CD3 (145-2C11, #100328, 1:200), APC-anti-CD4 (GK1.5, #100412, 1:200), PE-anti-CD4 (GK1.5, #100408, 1:200), APC-Cy7-anti-CD8α (53-6.7, #100714, 1:200), FITC-anti-CD25 (3C7, #101908, 1:200), PE-anti-CD40 (3/23, #124610, 1:200), FITC-anti-CD80 (16-10A1, #104706, 1:200), APC-Cy7-anti-CD86(GL1, #105030, 1:200), PerCP/Cy5.5-anti-I-A/I-E (M5/114.15.2, #107626, 1:200), FITC-anti-CD45.1 (A20, #110705, 1:200), APC-anti-CD326 (G8.8, #118214, 1:200), FITC-anti-H-2K^d (SF1-1.1, #116606, 1:200), PE-anti-IFN-γ (XMG1.2, 505808, 1:200), APC-anti-TNF-α (MP6-XT22, #506308, 1:200), FITC-anti-IL-17A (TC11-18H10.1, #506908, 1:200), APC-anti-CD11c (N418, #117310, 1:200), PE-anti-CD11b (M1/70, 101208, 1:200), PerCP/Cy5.5-anti-Ly6G (1A8, #127615, 1:200), APC-anti-Ly6C (HK1.4, #128015, 1:200), APC-anti-F4/80 (BM8, #13116, 1:200), 7-AAD (420404, 1:100). FOXP3 (FJK-16s, #12-5773-82, 1:200) was purchased from eBiosciences. Phospho-p44/42 MAPK (Erk1/2) (Thr202/Tyr204, #4370s, 1:800) were purchased from Cell Signaling Technology and Donkey anti-rabbit IgG-PE (Bio-Legend, San Diego, CA) was used as a secondary antibody.

For immunophenotyping, single-cell suspensions were obtained from intestines and spleen. To control for nonspecific binding, cells were blocked with anti-CD16/CD32 antibody (BD Biosciences, 2.4G2) for 10 min at room temperature. Surface staining for flow analysis was performed with 0.5 µL of antibody/test for 15 min at 4 ℃, protected from light. Cells were fixed and any red blood cells were lysed with Fix/Lyse solution (BD Biosciences) according to the manufacturer's instructions. For intracellular cytokine staining, T cells were stimulated with PMA/ionomycin (eBioscience) and treated with brefeldin A (eBioscience) for 5 h at 37 ℃. Permeabilization buffer (eBioscience) was used for intracellular cytokine staining. Intracellular cytokine staining was performed with 0.5 µL antibody/test for 30 min at room temperature in the dark. For IECs, cells were fixed with Fix/Lyse solution and permeabilized with Perm Buffer II (BD Biosciences) followed by antibody staining according to each manufacturer's protocol. Cells were run on an Accuri C6 or Attune NxT flow cytometer. Analysis was performed using FlowJo v10.2.

**Cytokine enzyme-linked immunosorbent assay (ELISA).** Serum from mice post-BMT or supernatants from cell culture were harvested and analyzed for IFN-γ (BD Biosciences), TNF-α (BD Biosciences), IL-6 (BD Biosciences), IL-1β (BD Biosciences), IL-17A (BioLegend), IL-18 (MBL), IL-22 (R&D systems, Minneapolis,

MN), and limulus amebocyte lysate assay QCL-1000 (Lonza) were performed following each manufacturer's instructions.

**Bacterial DNA sequencing and microbiota analysis.** On the indicated days, fecal pellets were collected from mice and stored at −20 ℃. DNA was isolated from fecal samples with a PowerMag Microbiome RNA/DNA Isolation Kit (Mo Bio Laboratories, Inc.) using an epMotion 5075 liquid handling system. The V4 region of the 16S rRNA gene was amplified and sequenced as described previously[47]. The 16S rRNA gene sequence data was processed and analyzed using the software package Mothur (v.1.38.1 and 1.39.1) and the most recent MiSeq SOP[48,49]. After sequence processing and alignment to the SILVA reference alignment (release 119)[50,51], sequences were binned into operational taxonomic units (OTUs) based on 97% sequence similarity using the average neighbor method[52,53]. By calculating θYC distances (a metric that takes relative abundances of both shared and non-shared OTUs into account)[54] between communities and using analysis of molecular variance (AMOVA)[55], it was possible to determine if there were statistically significant differences between the microbiota of different groups. Principle coordinates analysis (PCoA) was used to visualize the θYC distances between samples. Linear discriminant analysis (LDA) effect size (LEfSe) was used to determine if specific (OTUs) were differentially abundant in different groups[56] The taxonomic composition of the bacterial communities were analyzed by classifying sequences within Mothur using a modified version of the Ribosomal Database Project (RDP) training set (version14)[57,58]. For stool specimen analysis performed at Memorial Sloan Kettering Cancer Center, procedures were described previously[18].

**BMDC culture and isolation.** To obtain BMDCs, BM cells from WT-B6 or Gpr43^{−/−} mice were cultured with murine recombinant GM-CSF (20 ng/ml; PeproTech Inc., Rocky Mill, NJ) for 7 days and harvested[59]. BMDCs were then isolated using CD11c MicroBeads and an autoMACS (Miltenyi Biotec) resulting in purity >90%. Isolated BMDCs were then stimulated with LPS (0.5 µg/mL, Invivogen) for 16 h.

**Mixed lymphocyte reaction.** Splenic T cells from WT-B6 and BALB/c animals were used as responders, and WT-B6 versus Gpr43^{−/−} mice derived BMDCs were used as stimulators in a mixed lymphocyte reaction. $1 \times 10^5$ T cells and irradiated (20 Gy) $2.5 \times 10^3$ BMDCs were co-cultured on 96-well round-bottom plates for 72 and 96 h. The incorporation of ³H-thymidine (1µCi/well) by proliferating T cells during the final 16 h of co-culture was measured by a Betaplate reader (Wallad, Turku, Finland). When using splenocytes as stimulators, $4 \times 10^5$ T cells from WT-BALB/c animals and $1 \times 10^5$ irradiated (30 Gy) red blood cell lysed splenocytes from WT-B6 or Gpr43^{−/−} animals were co-cultured on 96-well flat-bottom plates for 72 and 96 h and analyzed for proliferation following ³H-thymidine incorporation during last 6 h of incubation. When using peritoneal macrophages or as stimulators, a total of $1 \times 10^5$ T cells and $1 \times 10^5$ isolated macrophages from [WT B6 → WT B6] and [WT B6 → Gpr43^{−/−}] animals were cocultured in 96-well U-bottom plates for 72 h. The incorporation of ³H-thymidine (1 µCi/well) by proliferating T cells during the final 6 h of coculture was measured.

**Non-specific TCR stimulation.** Isolated T cells ($1 \times 10^5$/well) were stimulated with anti-CD3 (2 µg/ml, 154-2C11, Biolegend) and anti-CD28 (1 µg/ml, 37.51, Biolegend) antibodies on 96 well round-bottom plates for the indicated periods. The incorporation of ³H-thymidine (1 µCi/well) by proliferating T cells during the final 6 h of culture was measured by a Betaplate reader (Wallad).

**Treg suppression assay.** CD4^+CD25^− and CD4^+CD25^+ T cells were isolated from spleens from WT-B6 or Gpr43^{−/−} animals using the CD4^+CD25^+ regulatory T cell isolation kit (Miltenyi Biotec) according to manufacturer's protocol. The purity of each type of cells was >90%. CD4^+CD25^+ T cells were serially diluted from $1 \times 10^5$ to 6250 cells/well and incubated with $1 \times 10^5$ CD4^+CD25^− T cells and $5 \times 10^5$ irradiated BALB/c splenocytes for 96 h. Incorporation of ³H-thymidine (1 µCi/well) by proliferating cells was measured during the last 16 h of culture.

**Organ explant culture.** To measure cytokine production, colon and ileum sections (1 cm) were dissected from WT B6 and Gpr43^{−/−} mice, washed in PBS, and then incubated overnight in 2 ml RPMI-1640 supplemented with 10% fetal calf serum, penicillin, and streptomycin. Tissues were treated with GLPG0974 (GPR43 antagonist, 1 µM, Tocris), sodium propionate and sodium butyrate (0.15, 1, 1.5 mM, Sigma), LPS (0.1 µg per mL, Invivogen), and U0126 (ERK inhibitor, 10 µM, Tocris) as indicated. Supernatants were collected and assayed for cytokine levels by ELISA. To stimulate NLRP3, tissues were incubated with 150 µg/ml of monosodium urate crystals (MSU, InvivoGen).

**Immunohistochemistry.** Tissues were processed, embedded in paraffin, and cut into 5 µm sections. Slides were de-paraffinized, and heat-induced antigen retrieval was performed with 10 mM sodium citrate buffer. Endogenous peroxidases were quenched with 3.0% hydrogen peroxide for 15 min. Primary anti-IL-18 antibody (BioVision, Milpitas, CA) was diluted 1:500 in PBST containing 10% goat serum (Thermo Fisher Scientific) and incubated for 60 min at room temperature.

Bound anti-body was detected using an anti-rabbit HRP labeled polymer incubated for 30 min (EnVision + system) and ImmPACT DAB (VECTOR laboratories, Burlingame, CA). Slides were then counterstained with hematoxylin, dehydrated, and covered.

**FITC-dextran assay**. Food and water were withheld from all mice for 4 h on day +14. FITC-dextran (Sigma-Aldrich) was administered by intra-gastric gavage needle at a concentration of 50 mg/ml in PBS. BMT recipients received 800 mg per kg (~16 mg per mouse). Four hours later, serum was collected from peripheral blood, diluted 1:1 with PBS, and analyzed on a plate reader at an excitation/emission wavelength of 485 nm/535 nm. Concentrations of FITC-dextran experimental samples were determined on the basis of a standard curve.

**Statistical analysis**. Bars and error bars represent the mean and s.e.m., respectively. We performed non-survival analysis using two-tailed $t$-test for statistical comparisons. Equality of variance between groups analyzed by an unpaired $t$-test was assessed with an $F$ test. The relative mRNA or protein expression levels were analyzed with two-tailed Mann–Whitney $U$ test. Statistical significance was determined with $a = 0.05$. We performed survival data analysis using a Mantel–Cox log-rank test. All statistical analyses were performed using GraphPad Prism 7. Samples sizes were estimated based on preliminary experiments. No statistical method was used to predetermine sample size.

## Data availability

The raw sequencing reads have been deposited at the NCBI Short Read Archive under BioProject ID PRJNA483178 and PRJNA482615. All other data are available from the authors upon request.

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

## Acknowledgements

This work was supported by the US National Institutes of Health grants HL090775, CA173878, CA203542 (P.R.), CA-039542 (J.L.M.F.) and JSPS Postdoctoral Fellowships for Research Abroad (H.F.) and The YASUDA Medical Foundation Grants for Research Abroad (H.F.). We acknowledge use of the Microscopy & Image-analysis Laboratory (MIL) of the University of Michigan's Biomedical Research Core Facilities for preparation of samples and images. Support for the MIL core is provided by the University of Michigan Cancer Center (NIH grant CA46592). This research was supported by work performed by The University of Michigan Microbial Systems Molecular Biology Laboratory.

## Author contributions

H.F. designed and performed experiments, analyzed the data, and wrote the paper. M.D.D. performed experiments, analyzed data, and wrote the paper. M.R. performed experiments, analyzed data, and edited the paper. D.P. performed experiments and edited the paper. T.T., I.H., S.J.W., S.K., A.T., S.B., C.L., C.Z., K.O.-W., and Y.S., performed experiments. C.L. performed experiments and histopathological analysis. G.N. provided mice and wrote the paper. J.E.L., J.L.M.F., and M.v.d.B. designed experiments and edited the paper. P.R. designed experiments, analyzed the data, and wrote the paper.
