## [Peer Review File · Nature Communications]

Reviewers' comments:

Reviewer #1 (Remarks to the Author):

The manuscript by Reddy and Colleagues shows a role for GPR43 in protection for experimental GVHD, and the metabolites butyrate and propionate.

The manuscript contains an enormous amount of experimental data, which was well performed and clear. The story is an extension of the authors previous paper in Nature Immunology, but here they provide clearer molecular mechanism.

I had no major suggestions or criticism of the experiments, and feel the authors have done enough and don't need to do months worth of extra experiments to fill some critical gap.

However the manuscript can be improved, particularly the discussion which I found uninspiring. I think the authors should include speculation on the following points-

- should gut dysbiosis/leaky gut be a major predictor for human GVHD
- Is there something intrinsic about GPR43 or will any measure that improves epithelial integrity mitigate GVHD
- Is there any clinical evidence for association of GVHD and lifestyle ie western vs Mediterranean diet etc
- Is gut leakiness the main action of poor GPR43 signaling? If so authors should somehow incorporate this in their summary figure, and possibly suggest effects of bacteria/LPS etc in the cellular and molecular mechanisms of exacerbated GVHD. Ie exactly what is happening? Dysbiosis through poor GPR43 signaling= leaky gut= passage of LPS etc to blood and tissues= stimulation of DC/macrophages and excessive T cell stimulation?
- GPR109a plays an equally important role in gut integrity, at least in DSS colitis. This could be mentioned as well as likely butyrate effect through this receptor

It is ok to say a result is difficult to explain, rather than brush it under the carpet. The fact that acetate (as high affinity for GPR43 as butyrate and propionate) had little effect if most of the mechanism is through GPR43. Statement that lots of acetate in the GI tract hard to reconcile. I would say that this is difficult to explain currently. Especially since acetate a metabolite for gut integrity (Ohno Nature, Marino Nature Immunology).

The authors did not measure IL-22, a critical cytokine for gut homeostasis. If this is easy (pre-existing samples) it should be done. It is another mechanism over inflammasome activation for gut homeostasis.

Reviewer #2 (Remarks to the Author):

The G-protein-coupled receptor 43 (GPR43), also known as FFAR2 is widely expressed and has been demonstrated to be important in a colitis model. Herein, the investigators suggest that it is important for attenuation of gastrointestinal GVHD and that GPR43 was critical for the protective effects of the short-chain fatty acids (SCFAs). The difference in GVHD in the absence of GPR43 was not due differences in the gut microbiota. The protective effect of SCFAs required GPR43 mediated ERK phosphorylation and activation of the NLRP3 inflammasome in non-hematopoietic target tissues of the host. This laboratory had already demonstrated that there is a protective effect of SCFA from GVHD and in this manuscript describe the results from manipulating the receptor side of SCFA, namely GPR43 in the intestinal epithelial cells (IEC).

1. The manuscript is well written and the experiments are logical and systematic. In one sense, it

is not surprising that since SCFA have been shown to be important, the receptor side for SCFA, namely GPR43 should be equally important.

2. Figure 1 shows a drop in GPR43 but this reviewer believes that by day 14 from an allogeneic transplant that the total number of cells are also significantly decreased. Are these data normalized by total number of IEC cells (not just beta actin).

3. It is interesting that GPR43 can heterodimerize with GPR41 (FFAR3) and signal downstream through beta arrestin, but in this model only GPR43 is affected? Is there a reason for selectivity for this particular SCFA sensor only?

4. The protection from GVHD appears to be a markedly faster death in the KO animals. Since these are global KO, is the effect not so much on the IEC side but also on the immune effector cells which also express GPR43?

5. The studies on the gut microbiome are of interest especially since in this model system there is no evidence (contrary to what this group has published before) that it plays a role in the regulation of GVHD. It is somewhat surprising given that the effects of SCFA are pretty clear. It is not clear to this reviewer the absolute value of these studies since the KO animals grow up without this receptor which is likely important in the selection of the original host microbiome. This altered microbiome likely impacts how the immune system "grows up."

6. It would be helpful for the authors to comment on the magnitude of improvement. For example, there is no doubt that the use of an agonist to GPR43 improves survival of the animals, but the protection and improvement appears to simply be a delay in survival with the majority of the animals succumbing by day 60. Therefore, the protective effects appears to be limited and mostly a shift in the timing of the mortality.

7. In Figure 4, it would have been helpful to inquire whether other cell types such as neutrophils and other myeloid cells (both at baseline and early after transplantation and at the time of recovery) are different in terms of numbers and function.

8. Do the bone marrow chimeras, namely WT B6 → GPR43^{-/-} have an inflammatory phenotype at baseline and therefore are more susceptible to GVHD?

We wish to thank the Editorial Board and the Reviewers for the thorough appraisal of our manuscript and for giving us the opportunity to submit a revised version of our manuscript. We are grateful for the thoughtful and constructive comments which we believe have enhanced our manuscript. We have in response performed additional experiments and now provide several additional datasets that answer **all** of the comments.

Please find below point by point responses to all of the comments. All of the changes are shown highlighted in the main manuscript.

Comments by Editor (E):

Comment E 1: In line with the reviewers we agree that the discussion requires extension, more detail regarding the magnitude of improvement during GPR43 agonist therapy is required.

Response: To determine the magnitude of improvement by GPR43 agonist, we have now performed additional experiments wherein we increased the severity of GVHD by increasing the dose of donor T cells. The greater the severity of GVHD, the longer the duration of agonist therapy was required. Specifically, the protection from mortality was statistically significant in more intense GVHD only when the agonist was administered for a longer duration. These results and discussion are now shown in the Results section on page 8, paragraph 2 and on page 9, paragraph 1, as and as new Supplementary Figures, Figure S2A-B.

Comment E 2:whether cell number reductions were due to a global reduction in cells should be addressed.

Response: We have normalized to the total actin. But we have also normalized for total IEC cell numbers at multiple time points after BMT between the groups, namely on days 7, 14 and 21. The IEC cell numbers for days 7, 14 and 21 are now provided as supplementary Figure S1A and Figure S1B and in Results section on page 5, paragraph 1.

Reviewer #1 (Remarks to the Author):

Comment 1.1: The manuscript contains an enormous amount of experimental data, which was well performed and clear. I had no major suggestions or criticism of the experiments, and feel the authors have done enough and don't need to do months' worth of extra experiments to fill some critical gap.

Response: We are grateful for the comment.

Comment 1.2: However the manuscript can be improved, particularly the discussion speculation on the following points.....

Response: We appreciate the thoughtful suggestion on discussing the scope of our study observation. As recommended we have now added an entire page to the discussion section to speculate and clarify on several of the points raised by the reviewer. The expanded discussion is now shown in the Discussion section on page 17 and on page 18 (paragraph 1).

Comment 1.3: ...If so authors should somehow incorporate this in their summary figure, and possibly suggest effects of bacteria/LPS etc in the cellular and molecular mechanisms of exacerbated GVHD.

Response: We agree. We have additional experiments that demonstrate a greater leakage of LPS into systemic circulation and a greater loss of intestinal barrier function by FITC dextran studies. These data are now shown in the Results section on page 12, paragraph 2 and as new Figures 4O and 4P. We have also, as suggested, revised the final mechanism cartoon Figure 7 to include LPS leakage and barrier breach.

Comment 1.4: GPR109a plays an equally important role in gut integrity, at least in DSS colitis. This could be mentioned as well as likely butyrate effect through this receptor.

Response: We now include this in the expanded discussion on page 17. Please also see above response to comment 1.2.

Comment 1.5: It is ok to say a result is difficult to explain, rather than brush it under the carpet. The fact that acetate (as high affinity for GPR43 as butyrate and propionate) had little effect if most of the mechanism is through GPR43. Statement that lots of acetate in the GI tract hard to reconcile. I would say that this is difficult to explain currently. Especially since acetate a metabolite for gut integrity (Ohno Nature, Marino Nature Immunology).

Response: We now acknowledge this on page 17 in the discussion section. Please also see above response to comment 1.2.

Comment 1.6: The authors did not measure IL-22, a critical cytokine for gut homeostasis. If this is easy (pre-existing samples) it should be done.....

Response: We have performed these experiments as recommended. The data are now provided as Figure 5F and as Supplemental Figure S4A and discussed in the Results section on page 13, paragraph 2.

Reviewer #2

Comment 2.1: The manuscript is well written and the experiments are logical and systematic. In one sense, it is not surprising that since SCFA have been shown to be important, the receptor side for SCFA, namely GPR43 should be equally important.

Response: We are appreciate and agree with the comments

Comment 2.2: Figure 1 shows a drop in GPR43 but this reviewer believes that by day 14 from an allogeneic transplant that the total number of cells are also significantly decreased. Are these data normalized by total number of IEC cells (not just beta actin).

Response: Please see response to Comment E2. We have normalized to the total actin. But we have also normalized for total IEC cell numbers at multiple time points after BMT between the groups, namely on days 7, 14 and 21. The IEC cell numbers for days 7, 14 and 21 are now provided as supplementary Figure S1A and Figure S1B and in Results section on page 5, paragraph 1.

Comment 2.3. It is interesting that GPR43 can heterodimerize with GPR41 (FFAR3) and signal downstream through beta arrestin, but in this model only GPR43 is affected? Is there a reason for selectivity for this particular SCFA sensor only?

Response: Our data show that only expression of GPR43 was altered after allo-BMT. We have now addressed the implication of this in the discussion section on page 17, paragraph 2. Please also see response to comment by reviewer 1, comment 1.2.

Comment 2.4. The protection from GVHD appears to be a markedly faster death in the KO animals. Since these are global KO, is the effect not so much on the IEC side but also on the immune effector cells which also express GPR43?

Response: The data from chimera experiments showed that allogeneic [WT B6→WT B6Ly5.2] and the [GPR43^{-/-}→WT B6Ly5.2] animals showed similar GVHD, but the WT B6 Ly5.2→GPR43^{-/-} chimeras showed significantly more severe GVHD indicating that effects was largely from expression on non-immune cells. However, in response (also as a response to comment 2.8), we have now performed additional studies and analyzed professional antigen presenting cells (macrophage and DCs) from the [WT B6 →WT B6] and [WT B6 →GPR43^{-/-}] animals after irradiation found no difference in the numbers of innate or T cell stimulatory functions of these cells. These data are now shown in supplementary Figure 3 as Figures S3G, S3H, S3I and in the Results section on page 12, paragraph 1. Collectively these data suggest that the dominant effect is from effect on non-immune cells, but we cannot formally rule all contribution from immune cells because of the limitations of the chimera studies.

Comment 2.5. The studies on the gut microbiome are of interest especially since in this model system there is no evidence (contrary to what this group has published before) that it plays a role in the regulation of GVHD. It is somewhat surprising given that the effects of SCFA are pretty clear. It is not clear to this reviewer the absolute value of these studies since the KO animals grow up without this receptor which is likely important in the selection of the original host microbiome. This altered microbiome likely impacts how the immune system “grows up.”

Response: We appreciate the thoughtful comment and apologize for lack of clarity. Our data suggests that one critical manner in which microbiome is important for GI GVHD is due to the metabolites it generates. Absence of GPR43 (host genotype) clearly sculpt the host microbiome (Fig 2A), but this change by itself is not sufficient to cause severe GVHD as demonstrated by our co-housing and mixing studies (Fig 2 and Supplementary Fig 1). But instead the inability of

these mice to respond to SCFAs in the absence of GPR43, even when administered exogenously, is critical for greater GVHD. Furthermore, the ex vivo immune cell responses (APCs and T cells) of the GPR43^{-/-} animals were comparable to WT animals. We agree that our data does not specifically address the contribution from the unique microbiome on how GPR43 contributes to the development of immune system or its contribution to GVHD, but within the limitations of the ex vivo studies, it does not appear to be distinct. The limitations and implication of our observations are further discussed in the Discussion section on page 17 and 18.

Comment 2.6. It would be helpful for the authors to comment on the magnitude of improvement. For example, there is no doubt that the use of an agonist to GPR43 improves survival of the animals, but the protection and improvement appears to simply be a delay in survival with the majority of the animals succumbing by day 60. Therefore, the protective effects appear to be limited and mostly a shift in the timing of the mortality.

Response: We agree and have now performed additional experiments to address this by increasing the severity of GVHD and then assess the magnitude of improvement with GPR43 agonists. These results are now shown as in the Results section on page 8, paragraph 2 and on page 9, paragraph 1, as and as new Supplementary Figures, Figure S2A-B. Please also see response to comment E 1.

Comment 2.7. In Figure 4, it would have been helpful to inquire whether other cell types such as neutrophils and other myeloid cells (both at baseline and early after transplantation and at the time of recovery) are different in terms of numbers and function.

Response: We appreciate the thoughtful comment. We have now performed these experiments and the impact on myeloid cells at baseline and on days 7 and 14, in the spleen, colon and small intestines after BMT is now shown in the Results section on page 10, paragraph 2 and in supplementary Figure 3 as Figure S3D.

Comment 2.8. Do the bone marrow chimeras, namely WT B6 → GPR43^{-/-} have an inflammatory phenotype at baseline and therefore are more susceptible to GVHD?

Response: We have now analyzed the professional APCs, macrophages and DCs, from these chimeras and determined their phenotype and inflammatory responses at baseline and found no significant increase in numbers or functions. These results are now presented in the Results section on page 12, paragraph 1 and as supplementary Figures as Figure S3G, S3H and S3I.

We now hope our manuscript will be acceptable.

Thank you for your kind consideration.

Sincerely,

Pavan Reddy

REVIEWERS' COMMENTS:

Reviewer #1 (Remarks to the Author):

the manuscript is improved. typos in the text. Note GPR109a not GPR101

Reviewer #2 (Remarks to the Author):

all concerns were addressed

Reviewer #1: The manuscript is improved. Typos in the text. Note GPR109a not GPR101.

Response: Thank you. We have corrected the typos in the text. We have made the relevant changes to the accurately reflect it as GPR109a in the discussion:

Page 17, line 2 (Discussion): GPR101 to GPR109a

Page 17, line 14 (Discussion): GPR101 to GPR109a

Page 17, line 23 (Discussion): GPR101 to GPR109a

Reviewer #2: All concerns were addressed

Response: Thank you and very appreciative.